

# Characterizing drought by change in precipitation-runoff relationship: a case study of the Loess Plateau, China

Yuan Zhang[1,2], Xiaoming Feng[1], Xiaofeng Wang[2], Bojie Fu[1]

[1] State Key Laboratory of Urban and Regional Ecology, Research Center for Eco-Environmental Sciences, Chinese Academy of Sciences, Beijing 10085, China

[2] School of Earth Sciences and Resources , Chang'an University, Xi'an, Shaanxi 710054, China

*Correspondence to*: Xiaoming Feng (fengxm@rcees.ac.cn)

**Abstract.** The frequency and intensity of drought is increasing dramatically with global warming. Yet, few studies have characterized drought from its impact on the ecosystem services, the mechanisms through which ecosystems support life. As a result, little is known about the implications of increased drought on resource management. This case study characterizes drought by linking climate anomaly with the change in precipitation-runoff relationships, in the Loess Plateau of China, a

water-limited region where re-vegetation in the area makes drought a major concern. We analyze droughts with duration greater than 5 years and annual precipitation anomalies more negative than -5%, we found that continuous precipitation shifts is able to change watershed water balance in the water limited area, multi-year drought caused the precipitation-runoff relationship to change with a significantly descending trend (p<0.05) compared to other historical records. For the whole Loess Plateau, the average runoff ratio decreased from 10 percent to 6.8 percent in 1991–1999. The joint probability and return period

gradually increased with the increase of drought duration and severity. The ecosystem service of water yield was affected easily when the drought duration is not less than six years and the drought severity is greater than or equal to 0.55 (precipitation around ≤212mm). At the same time, the growth ratio of annual NPP is also susceptible to prolonged drought, the growth ratio is lower in these watersheds which had a significant change in the PRR. Such studies are essential to ecosystem management in a water-limited area.

## 1 Introduction

Drought is a complex and recurrent climatic phenomenon, causing far-reaching and negative impacts on agriculture, water resources, the environment and human life (Ghulam et al., 2007; Goddard et al., 2001). Few extreme events are as economically and ecologically disruptive as drought over the past half a century (Dai, 2011). For example, multi-year droughts across the globe have triggered an increase in tress mortality which is linked to climate change (Allen et al., 2010). At

regional scale, during 1998–2001, one-third of Iran's territory was affected by the most severe drought in the country's history with more than half of the country's population facing food shortages and insufficient drinking-water supplies(Raziei et al., 2009). The annual economic losses caused by droughts in the United States are estimated as being up to US$6-$8 billion (Wilhite, 2000). As a result of these impacts , an increasing number of studies are now focusing on characterizing drought



events, including drought identification and frequency analysis, and the necessary resource management actions drought bring (Michele et al., 2013).

Many drought indices have been developed to monitor drought evolution on region and global scales (Yan et al., 2016).

Meteorological drought can be identified by sample anomalies in precipitation data, such as the Palmer Drought Severity Index(PDSI) (Palmer, 1965), Standardized Precipitation Index(SPI) (Mckee et al., 1993), Precipitation Anomaly (Moron, 1994) and the Bhalme-Mooley Index (Bhalme and Mooley, 2009). Others recognize the necessity of developing drought indicators that reflect the causes and impacts of drought, these definitions of drought index incorporate many different physical, biological and socioeconomic variables (Sheffield et al., 2004). Examples as: the soil moisture indicator (Xia et al., 2014), crop

drought indicator (Duff et al., 1997) and the crop water demand indicators that are used to reflect the decreased harvests of crops suffering water stress under drought conditions. For hydrological-drought indicators (Herbst et al., 1966; MOHAN and RANGACHARYA, 1991), droughts are usually identified based by run-length theory and are characterized as periods where water demand exceeds supply because of long term precipitation shortage. In addition, socioeconomic drought can reflect the undesirable social and economic impacts induced by the above-mentioned drought types. All these drought indicators are

based on defining a single variable to quantify a drought event. However, drought conditions are associated with multiple variables, a single drought indicator can not be sufficient to characterize the complicated drought condition and reflect its wide impacts (Hao and Singh, 2015) At the same time, ecosystem services are closely related to our lives (Burkhard et al., 2014; Zheng et al., 2016), these drought indicators can also not reflect its impact on ecosystem and remain less informative to the policy maker and resource management.

The precipitation-runoff relationship (PPR) is an important issue in engineering hydrology, water resource planning and management, and watershed system evolution (Guo et al., 2016; Nourani et al., 2015). The focus of current researches involving the change and transformation process of the precipitation-runoff relationship. Charlier et al. (2015) applied the wavelet transform to detect the change of PPR. Sun et al. (2016) assessed hydrologic trends in urban catchments by using

conceptual urban precipitation-runoff model, which is beneficial to the stormwater management and planning. There is no doubt that the variation in the PPR reflects the actual integrated volume of precipitation and runoff. The ratio between the annual runoff and the annual precipitation, the so-called "runoff ratio" (Savenije, 1996; Feng et al., 2016b), can usually be used to quantify the ecosystem service of water yield. This ecosystem service is of major concern in water-limited areas because it represents the water resource available to human beings. Studies have shown that protracted drought may affect

runoff generation and cause changes in the PPR (Petrone et al., 2010; Saft et al., 2015). So the shift in PPR caused with an extended drought will eventually have an adverse effect on the ecosystem service of water yield. Based on this perspective,



we want to characterize drought by change in the PRR and investigate the characteristics and frequency of these dry events, thus further optimize ecosystem management.

5 Drought stress is the main environmental factor limiting terrestrial ecosystem productivity in arid and semi-arid areas (Boyer, 1982). In China's Loess Plateau, a typical human and nature coupled system, the ecosystem service of water yield is of concern not only to the sustainability of the re-vegetation program but also related to the objective of raising the level of human-economic development (Wang et al., 2015; Feng et al., 2016a). As one of the hydrological services, water yield plays a vital role in developing human society while maintaining ecological security. In recent years, the Loess Plateau climate has been warming and drying (Lü et al., 2014) as the global temperature increases, a change that will eventually affect the balance 10 between regional water supply and demand. So analyzing drought characteristics based on the response of the PRR change to multi-year dry periods is of great importance in estimating reflect the effect of drought and the ecological re-construction of the whole Loess Plateau.

The objectives of our study are to characterize drought in the Loess Plateau. We first link climate and the PRR to define 15 drought; we then simulated the drought frequency and return period of different drought events. Finally, we discuss the policy implications of our study to ecosystem management.

## 2. Materials and Methods

### 2.1. Study area

The Loess Plateau(Fig. 1) is located between 100°54′E-114°33′E and 33°43′N-41°16′N, covering a total area of 624,000 km$^2$. 20 It has a semi-arid and semi-humid climate with annual precipitation ranging from 200 mm in the northwest to 700 mm in the southeast. It is a key area in the current and past efforts to conserve soil and water in the Yellow River basin. The ecology of the Loess Plateau is sensitive to climate change.

In this study, we use the 13 watersheds, which comprise the main watershed of the Loess Plateau(Fig. 1). River runoff from 25 these thirteen watersheds (about 35% of the Loess Plateau area) contributes 65% of the discharge into the middle reach of the Yellow River. The attributes of each basin are shown in Table 1. Runoff and meteorological data for the 13 watersheds were obtained from the Yellow River Conservancy Commission (http://tghl.forestry.gov.cn/) and National Meteorological Information Center (NMIC; http://cdc.nmic.cn/home.do) of the Chinese Meteorological Administration(CMA) respectively. Some studies have shown that the Grain for Green Program(GGP) began in 1999 resulted in the reduction of runoff in Loess 30 Plateau (Zhang et al., 2008; Feng et al., 2016b), so the study period is selected from 1961 to 1999.





### 2.2. Drought identification

We first calculated the precipitation anomalies in the whole Loess Plateau its separate watersheds, the anomaly series were divided by the mean annual precipitation and smoothed with a 3-year moving average. The basic rules for identifying drought events are (1) A single year's precipitation anomaly more negative than -10%; (2) The precipitation anomaly negative 0 for

more than three consecutive years.

The precipitation anomaly of drought events with condition (1) or cumulative value of (2) is taken as the corresponding drought severity (for convenience, drought severity is multiplied by -1 to obtain a positive value). Based on the rules above-mentioned, we recognized all dry events in every watershed. There is a limit to the duration and severity of drought

events: this is to reflect the response of PRR to drought events over the years. The drought events studied here mainly had a duration of not less than five years and a mean annual precipitation anomaly more negative than -5% during the drought period. Then we classified these dry events as the main dry period, others as the single dry period.

We used the Kolmogorov-Smirnov (K-S) (Massey, 1951) test to determine whether annual precipitation and runoff data

followed an roughly normal distribution, if they did not, they were transformed with a Box-Cox transformation (Box and Cox, 1964). After identifying the main drought events, we tested whether the PRR change was statistically significant compared to the historic record using Student's T-test ($p \leq 0.05$).

### 2.3. Drought frequency analysis

Drought can be characterized by multiple variables, such as duration, severity, and spatial extent (Steinemann and Cavalcanti,

2006; Hayes et al., 2012), but how to determine the joint distribution between these variables has become an important issue. Here we propose use the copula function (Shiau, 2006). We chose two main factors affecting drought characteristics, drought duration and severity, to construct a joint distribution function. If the marginal distribution functions of drought duration($d$) and drought severity($s$) are $F_D(d)$ and $F_S(s)$ respectively, a copula C, exits that combines these two marginal distributions to give the joint distribution function, $F_{D,S}(d, s)$:

$$F_{D,S}(d,s) = C(F_D(d), F_S(s)) \tag{1}$$

If the marginal distributions $F_D(d)$ and $F_S(s)$ are continuous, $f_D(d)$ and $f_S(s)$ are the density functions corresponding to $F_D(d)$ and $F_S(s)$, respectively, then the joint probability density function becomes:

$$f_{D,S(d,s)} = c(F_D(d), F_S(s)) f_D(d) f_S(s) \tag{2}$$

where $c$ is the density function of $C$, which is defined as:




$$c(F_D(d), F_S(s)) = \frac{\partial^2 C(F_D(d), F_S(s))}{\partial F_D(d) \partial F_S(s)} \tag{3}$$

### 2.4. The criteria for deciding the parameters and goodness-of-fit test

#### 2.4.1. Parameters estimation

Maximum Likelihood(ML), Inference Functions for Margins(IFM) and Canonical Maximum Likelihood(CML) are
commonly used in parameter estimation (Mirabbasi et al., 2012; Lee et al., 2013). Here, we used ML and IFM to estimate the
parameters of the marginal distribution function and the joint copula functions of drought duration and severity, respectively.
We chose seven commonly used distributions to describe univariate probability distributions as the candidate margins for
drought duration and severity. They are the exponential, gamma, log-normal, extreme value, generalized extreme value,
Poisson and Weibull distributions. The K-S test was used to establish the optimal margin distribution function. By comparison,
drought duration and severity are fitted with Weibull and gamma distributions respectively and the Kendal correlation
coefficient of their empirical distribution and theoretical distribution function are tested by 0.05 significance T-test. Using ML
to estimate the distribution parameter of drought duration($\alpha_1, \beta_1$) and severity($\alpha_2, \beta_2$),and then calculating the parameter of
copula function by the following formula:

$$\ln L(d, s; \alpha_1, \beta_1, \alpha_2, \beta_2, \theta) = \ln L_C(F_D(d), F_S(s); \theta) + \ln L_D(d; \alpha_1, \beta_1) + \ln L_S(s; \alpha_2, \beta_2) \tag{4}$$

where $\ln L_C$ is the log-likelihood function of the copulas.

#### 2.4.2. The determination of the optimal joint distribution function

Five commonly used two-dimensional functions were constructed using the marginal distribution function of drought duration
and drought intensity (Table 2). The goodness-of-fit test was performed by calculating the Squared Euclidean Distance(SED)
between the theoretical copula and the empirical copula (Berg, 2009). The empirical copula and SED are defined as:

$$\hat{C}(u) = \frac{1}{n+1} \sum_{j=1}^{n} I\{Z_{j1} \leq u_1, \ldots, Z_{jd} \leq u_d\} \tag{5}$$

$$d^2 = \sum_{i=1}^{n} \left| \hat{C}(u_i, v_i) - C(u_i, v_i) \right|^2 \tag{6}$$

### 2.5. Drought return period

$N$ is drought series length and $n$ is the number of drought events. The return period of single variable can be obtained from the
copula function definition as:

$$T(d) = \frac{N}{n[1 - F(d)]} \tag{7}$$





$$T(s) = \frac{N}{n[1-F(s)]}$$

(8)

The joint distribution function of drought duration and severity is:

$$F(d,s) = P(D \le d, S \le s) = \int_{-\infty}^{s} \int_{-\infty}^{d} f(s,d) d_u d_v = C(F_D(d), F_S(s))$$

(9)

The joint return period of two characteristic variables is calculated as:

$$T_a(d,s) = \frac{N}{nP(D \ge d \bigcup S \ge s)} = \frac{N}{n(1-C(F_D(d),F_S(s)))}$$

(10)

## 3. Results

### 3.1. Drought event in the Loess Plateau

There are 17 years, out of the 39 years of the study period, when the precipitation anomaly is negative. Based on the drought

identification method developed in this study, 7 dry periods are identified (Fig. 2 including the main dry period and single dry

period). We found that 1968-1974, 1979-1983 and 1991-1999 were all droughts with the duration longer than five years having

corresponding drought severity up to 0.43, 0.41 and 0.61 (corresponding average precipitations of 268 mm, 277 mm and 183

mm), respectively.

The K-S test on precipitation-runoff data from 1961 to 1999 shows that the precipitation-runoff data in this time series

approximate to a normal distribution, providing the premise for a linear relationship between precipitation and runoff.

Analyzing the change of the PRR in three main periods of Loess Plateau, the results shows that the PRR may change

significantly during these drought periods (Fig. 3). During the drought period of 1968–1974 ($p$=0.758) and 1979–1983

($p$=0.514) no significant change was found although the dry period regression line deviates from the overall regression. In

1991–1999 ($p$=0.000) there was a significant decrease change significantly in the PRR. In this period, the dry period regression

lines are lower than nearly all the other points indicating unprecedentedly low runoff generation rates for the given

precipitation. The long-term average runoff ratio is 10 percent. In the dry period of 1991–1999, the average runoff ratio

decreased to 6.8 percent.

### 3.2. Drought frequency in the Loess Plateau

The goodness-of-fit test for the joint function was performed by calculating the squared Euclidean distance between the

theoretical copula and the empirical copula according to the criterion of the smaller the distance, the better the function fits the

data. It can be seen from Table 3 that the Frank-copula function has the smallest squared distance except for Gushan, Dali and

Weihe watersheds, indicating that the Frank-copula function is more suitable for fitting the duration and severity of the drought




events in the study area. Note, for those three watersheds mentioned above, the distance value of the Frank-copula function is also relatively small. Finally, the Frank-copula function is selected as the joint distribution function of the two characteristics of drought duration and severity.

It can be seen from Figs 4a and 4b that the joint probability increases with the increase of drought duration and drought severity, this occurs regardless of the joint cumulative probability three-dimensional or contour lines. When the drought severity is 0.1–0.35 (corresponding to precipitation from 306 mm–423 mm), the contours are intensive and the duration of drought varies greatly (ranging from 1 to 9 years). When the drought duration is less than five years, the cumulative probability of the bivariate increases rapidly with the increase in drought severity; in contrast, the increasing trend of the joint probability slows

down with the increase in drought severity when the drought duration last more than 5 years. When the drought severity is not more than 0.4 (precipitation ≥ 282 mm), the joint probability of drought increases rapidly with the increase of drought duration, and the increasing trend of the joint probability slows down with the increase in drought duration when drought severity is greater than 0.4 (precipitation＜282 mm). It can be seen from the density of the contours that when the drought events in the Loess Plateau lasted for 4–6 years and the drought severity was in the range of 0.3–0.5 (corresponding precipitation was 235

15 mm–329 mm) during 1961–1999. Joint probabilities of droughts are important for drought management. The probability that both the drought duration and the severity simultaneously exceed certain thresholds is useful information for environmental and government agencies responsible for water system management under drought conditions.

Figs 4c and 4d show that with the increase of drought duration and drought severity, the joint return period also shows an

20 increasing trend. During the study period, when the duration and severity of drought events reached the maximum, the joint return period of such drought events in the Loess Plateau was close to 22 years. For the three major drought events in the Loess Plateau from 1961 to 1999 the return period was about 7.29 years, while the drought duration was 7 years or drought severity up to 0.43 (1968–1974). The return period reached 5.85 years when drought occured in 1979–1983 with a duration of 5 years and a severity of 0.43. The most severe drought event in the study period was 18.31 years with duration of 9 years and a

25 severity of 0.61. It can be seen that for the drought event of 1968–1974, the drought period of 7 years or the drought severity up to 0.43 was about 8 years from the time commencement of the drought, and in 1979–1983, the drought severity reached 0.41, which was close to the predicted return period.

### 3.3. Variability of the PRR during dry period in each watershed

Prolonged multi-year drought has caused significant damages in the natural environments. Fig. 5 demonstrates the range of

30 changes in the PRR under sustained precipitation decrease. According to the direction of change, the dry period regression line is mainly located upward or downward the overall regression, the change of the PRR in the studied 13 watersheds show



no significant test when the regression line of the dry period was above the total regression line. In the 15 times dry events under the regression line in the case of the total regression, there are 9 times significant change in the PRR ($p < 0.05$), accounting for about 60% of the total situation. This means that sustained drought results in lowered runoff generation rates for similar precipitation amounts (Saft et al., 2015). Thus in a continued years with decreased precipitation, we can conclude

that lower runoff not only relate to the lower precipitation, but also less runoff than expected caused by the multi_yeat drought period.

There were no significant changes in the PRR in 5 of the 13 watersheds during the study period. Fig. 6 demonstrates that there is no geographical pattern in the spatial distribution of watersheds with and without significant change in the PRR. Compared

to the annual average precipitation in other basins where significant changes occurred (Table1), these watersheds where there were no significant changes in precipitation-runoff relationship (Kuye, Dali, Qingjian, Yanhe, Jinghe) had higher precipitation. Thus we conclude that the PRR has a strong response to multi-year drought period with the occurrence of drought for many years being more likely to cause a significant change in the PRR in the basins with less precipitation.

### 3.4. Spatial variability of drought frequency in the Loess Plateau

With the difference of drought duration and severity, the return period also showed different values. Therefore, the three major dry events in the Loess Plateau from 1961 to 1999 were selected as the basis to study the spatial distribution characteristics of the joint return period. It can be seen from Figs 7a,7b and 7c that although the duration and severity of drought in these three major drought events are different, the spatial distribution of drought return period is consistent. At the same time, there are also differences in drought return period in different watersheds, and the spatial heterogeneity characteristics of different

catchments in the Loess Plateau can also be seen from the drought return period. In terms of spatial distribution, the return period of drought events in the southern and eastern watersheds of Jinghe, Beiluo, Xinshui and Fenhe are longer, indicating that the frequency of drought events in these watersheds is relatively low, while those in the north and west, such as Huangfu, Kuye, Tuwei, Weihe have a shorter drought return period, that is, the frequency of drought events in this region is higher than other regions, and drought events are more likely to occur. Analyzing the return period corresponding to watersheds with

significant changes in the relationship between precipitation and runoff Fig. 7d, it can be seen that there are obvious differences in the return periods in these watersheds, which are affected by different drought characteristics.



## 4. Discussion

### 4.1. Reliability of the identified drought event

Since the actual occurrence of drought in a region is complicated, and the practical significance of various drought indicators is different, the choice of drought definition is an important part of studying the process of drought occurrence and development.

Prolonged multi-year drought has caused significant damages both in the natural environments as well as in the development of the human societies (Belal et al., 2014). To ensure that the dry periods are sufficiently long and severe, Saft et al. (2015) only using dry periods with drought duration ≥7 years and severity <-5%. In this study, the duration of drought was limited to not less than 5 years. Based on the response of the PRR to drought events over the years, the results of drought events in the Loess Plateau during 1961–1999 showed that 1962, 1965, 1986–1987 and 1989 are single dry periods, 1968–1974, 1979–1983 and

1991–1999 are the main dry periods. Looking at the dry events in the Yellow River basin from 1961 to 1999, the year when major dry events occurred in this period were 1965,1972,1980,1995 and 1997 (Fu et al., 2008; http://www.mwr.ov.cn). For example, a severe drought over northern China in 1997 damaged 1.94 million hectares of crops in the Yellow River and resulted in 226 days of zero flow from Henan to Shandong provinces, and the total length of the river with zero flow was about 687 km. We can find these dry events which are identified by the method in this study are consistent with the statistical data in

historical, further illustrate the feasibility of the method in this study.

### 4.2. The influence of multi-year drought on the ecosystem service of water yield

Ecosystem service is an important concept for policy makers, and the variability of the relationship between precipitation and runoff is vital in the study of the ecosystem service of water yield. There are two main forms of the response of the PRR to multi-year droughts: no significant change and downward significant change. Saft et al. (2015) have shown that there was also

a significant upward trend in the PRR during a drought period in southeastern Australian watersheds, but the probability of such events is small. It is also not clear whether the phenomenon is real or just sampling fluctuations. Analyzing the changes of the PRR during the three main dry periods in the Loess Plateau, we found that runoff reductions were smaller than other historical when the precipitation sustained reductions in 1968–1974, and reductions in runoff were slightly greater than other periods in the latter two dry events. It can be seen that the annual runoff tends to decrease gradually. So we believe that the

occurrence of dry events will aggravate the reduction of runoff, which could lead to significant change of the PRR. The increase of extreme temperature indices in the Loess Plateau, such as txq90 (hot-day threshold), tn90p (warm-night threshold) and txhw90 (longest heatwave) (Vincent and Mekis, 2006; Zhou and Ren, 2011), will cause the reduction of runoff since1960s (Li et al., 2010). At the same time, the land use of the Loess Plateau changed greatly before and after the 1990s (Zheng et al., 2009), and the increase in human activities has also affected the reduction of runoff (Feng et al., 2016b). Potter





et al. (2010) also attributes the significant reduction in the internal runoff of the Murray-Darling Basin to lower precipitation and the rise of temperature. This study shows that sustained precipitation changes also have the capacity to transform watershed water balance in water limed area.

Drought regressions fell under the total regression lines 15 times, including significant changes and non-significant changes in the study period. During the drought runoff was reduced expected when the precipitation sustained reductions, which will decrease the runoff ratio and affect the ecosystem service of water yield. Based on the analysis of the duration and severity of the 15 dry events (Fig. 8a), it is concluded that the occurrence of a drought event is more likely to result in a significant descending trend of the PRR when that the drought duration is not less than 6 years and the drought severity is greater than or

equal to 0.55 (precipitation≤212 mm). Containing less runoff than expected is a problem for ecosystem service of water yield, the shift in the PRR will induce a contradiction between runoff expectations and the actual amount of water in watersheds if the runoff fail to predict accurately when the PRR changes. This suggests we must concern the impact of prolonged drought on ecosystem and optimize the modeling techniques of precipitation-runoff to copy with longer term influences in response to changed climatic conditions.

**4.3. Policy implications to ecosystem management**

In order to solve the problem of serious ecosystem degradation in the Loess Plateau and to effectively control soil erosion, in 1999 the Chinese government began a large-scale project of returning farmland to forest and grassland. Based on the spatial distribution of drought events in the Loess plateau, the 13 watersheds were further divided into four regions. It can be seen that the drought return period in the northern and western regions is lower than that in the central and eastern regions, and we can

see the distribution of the effects of the project in the four different regions (Fig. 8b). The different spatial distributions of return period further reflect regional differences.

Drought can affect ecosystem productivity and reduce the carbon sink capacity of terrestrial ecosystems (Burton and Zak, 1998; Ciais et al., 2005; Tian et al., 1998). We used the terrestrial Carnegie-Ames-Stanford Approach (CASA) ecosystem

model to estimate ecosystem net primary productivity (NPP) from remotely sensed data, specific method refers to (Feng et al., 2013), the change in annual NPP showed an increasing trend during 2000–2008 across the Loess Plateau. Compared the average growth rate of annual NPP in watersheds between Fig. 9a and Fig. 9b, the results illustrate average growth rate of annual NPP in Fig 9b is about 1.2 times higher than those watersheds which have a significant change in the PRR. Since the responses of different vegetation types to drought events at different time scales are not the same, the regional drought return

period and vegetation type should be taken into account in future policy and appropriate policies should be formulated





according to the specific regional conditions, so as to control the adverse effects of drought on ecosystem productivity. We must pay more attention these drought characteristics what can induced the PRR had a significant change, combined the return period (Fig. 7d), thereby avoiding the waste of capital investment and improving the implementation of the GGP effectively.

The results of this study provide a basis for the guidance of ecosystem management policy. In addition, the specific measures for adapting to drought should also be improved. For example, drought-resistant crops such as millet and maizes should be chose, farmers can also increase the soil water storage capacity by reclaiming level terraces, contour strip farming (Panagos et al., 2015) and ridge-furrow cropping (Gan et al., 2013). At the same time, implementing water-saving irrigation technology, such as drip irrigation or micro-irrigation (Zou et al., 2013) would not only improve the utilization efficiency of water

resources, but also resolve the issue of soil salinization caused by flood irrigation. Vegetation can affect the conversion of surface energy, water, momentum and biochemicals through its physical and physiological processes, thus affecting the atmospheric conditions (Bonan, 2008), further changing the regional precipitation and hydrological processes (Ellison et al., 2012). In the afforestation, managers should select tree species with less water consumption, reduce the density of planting, and consider the optimal distribution pattern between woodland, shrub and grassland to cope with the effect of warmer and

dryer conditions. For example, adopting fish-scale pits (Wang et al., 2014) to enhance the infiltration of atmospheric precipitation and soil moisture. Especially in areas with steep slopes, the use of rainwater harvesting measures such as fish-scale pits will boost the survival of trees and avoid soil drying.

## 5. Conclusions

This article characterizes drought by linking climate anomaly with the change in precipitation-runoff relationships. We found

that multi-year drought caused the PRR to have a significant descending trend ($p < 0.05$) compared to the historical records. For the whole Loess Plateau, the average runoff ratio decreased to 6.8 percent of the average annual precipitation compared to 10 percent of the annual average precipitation in 1991–1999. In 9 of the 13 watersheds studied significant change in the PRR during 1961–1999. When we compared the annual average precipitation in separate watersheds and analyzed their drought characteristics, we concluded that this situation is likely to happen when the drought duration is not less than 6 years and the

drought severity is greater than or equal to 0.55 (annual precipitation≤212 mm).

Our analysis revealed great spatial variability in drought across the Loess Plateau. We chose the Frank-copula function as the optimal joint distribution function of drought duration and severity. The results demonstrated that the joint probability and return period gradually increased with the increase of drought duration and severity. At the same times, spatial heterogeneity

characteristics of different watersheds in the Loess Plateau can be seen from the spatial distribution of drought return period.

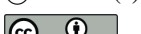

By analyzing the annual NPP over the Loess Plateau, we also found that pronged drought can also affect the growth ratio of NPP, the average annual growth ratio is lower in these watersheds which had a significant change in the PRR. Long-term drought is also an indispensable factor when considering the influencing of NPP trends in the future.

These results should lead to better water regulation and more effective strategies of ecosystem management. We can consider different plant species, combining spatial variability in drought events, to maximize the function of the ecosystem based on the stability of the ecosystem structure.

**Acknowledgments**

We acknowledgement the use of the runoff data sets available at Yellow River Conservancy Commission
(http://tghl.forestry.gov.cn/),climate data from National Meteorological Information Center (NMIC) of Chinese Meteorological Administration(CMA) ( http://cdc.nmic.cn/home.do). This work was funded by the National Natural Science Foundation of China (No. 41390464 and 41561134016), Key Basic Research of Ministry of Science and Technology (2016YFC0501603).

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





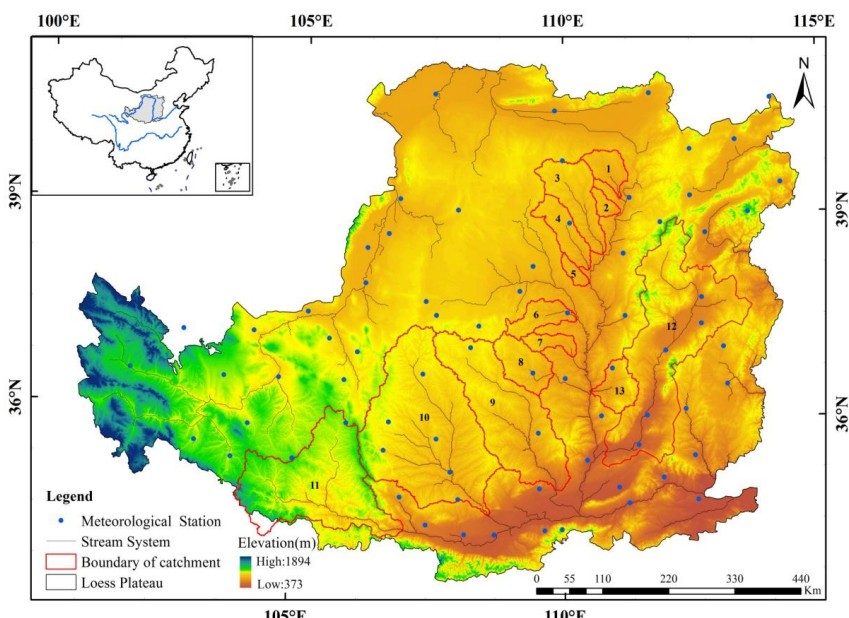

**Figure 1. The study area.**





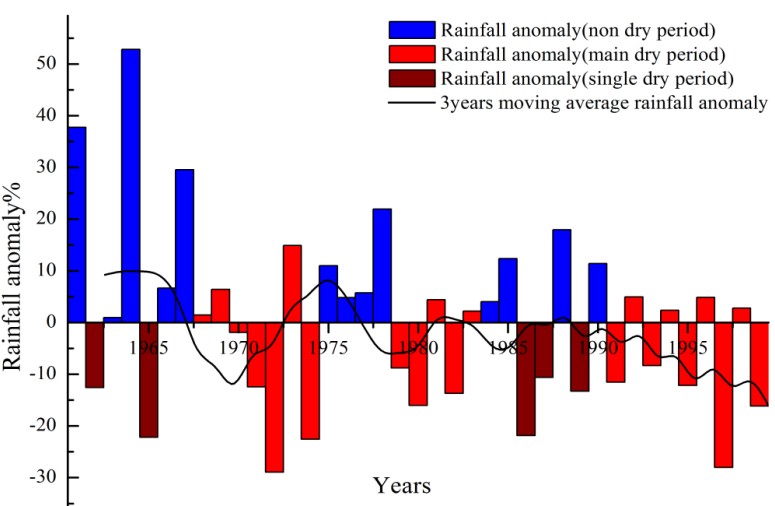

**Figure2. Time series with identified drought periods in the Loess Plateau during 1961-1999.**




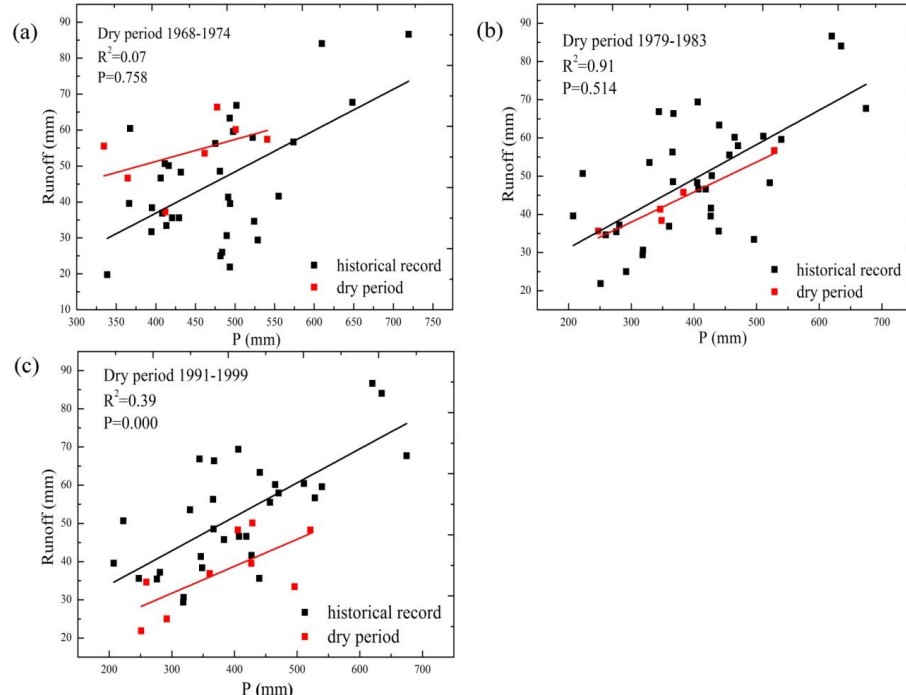

**Figure 3. Precipitation-runoff relationships during drought periods:(a,b) no significant change in precipitation-runoff relationship**

**and (c) significant downward change in precipitation-runoff relationship.**





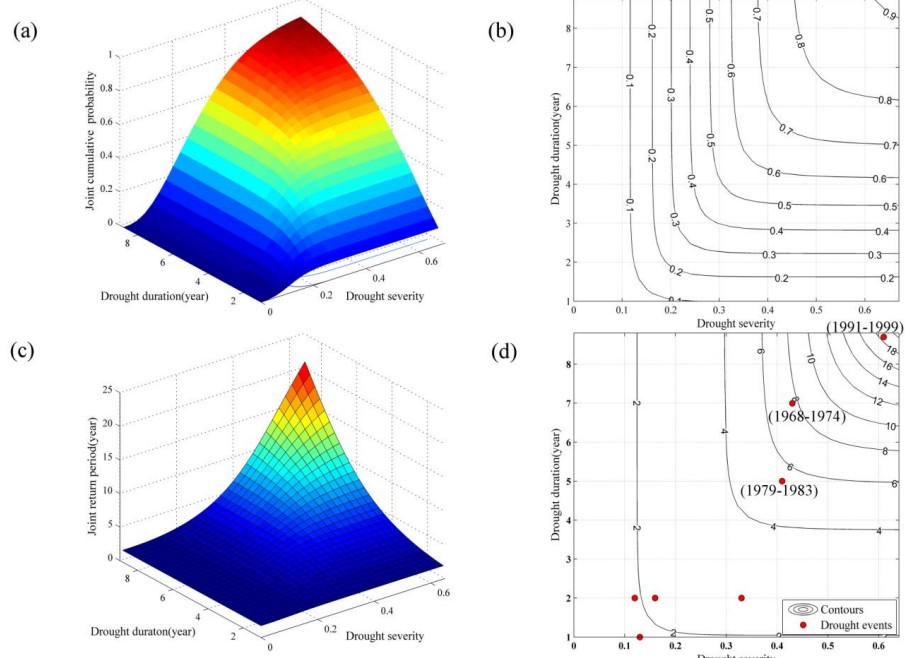

**Figure 4. The joint probability distribution of drought duration and drought severity and joint return period: (a) and (b) are the three-dimensional and contour maps of the joint probability density function of Frank-copula, respectively; (c) and (d) are the three-dimensional and contour maps of the joint return period of drought duration and drought severity, respectively.**





**Figure 5. The annual precipitation-runoff scatter plot in each watershed.**





Figure 5. (continued).



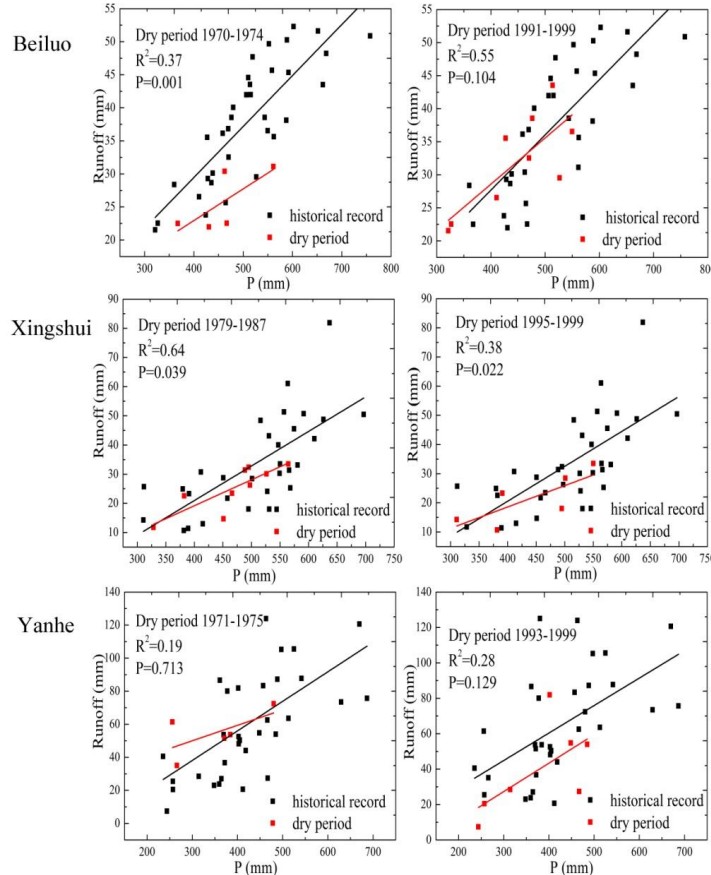

**Figure 5. (continued).**




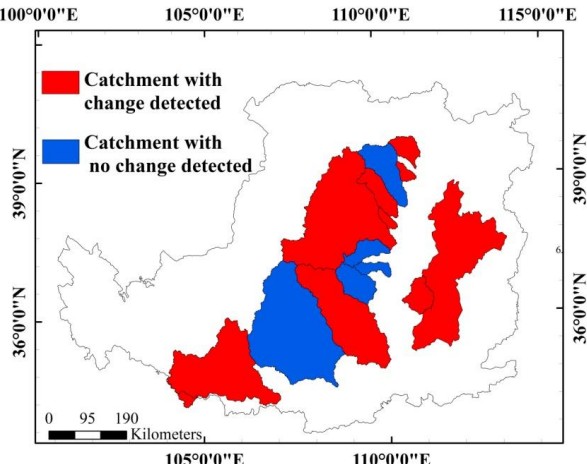

**Figure 6. Spatial distribution of watersheds with and without significant change in the PPR during drought period.**




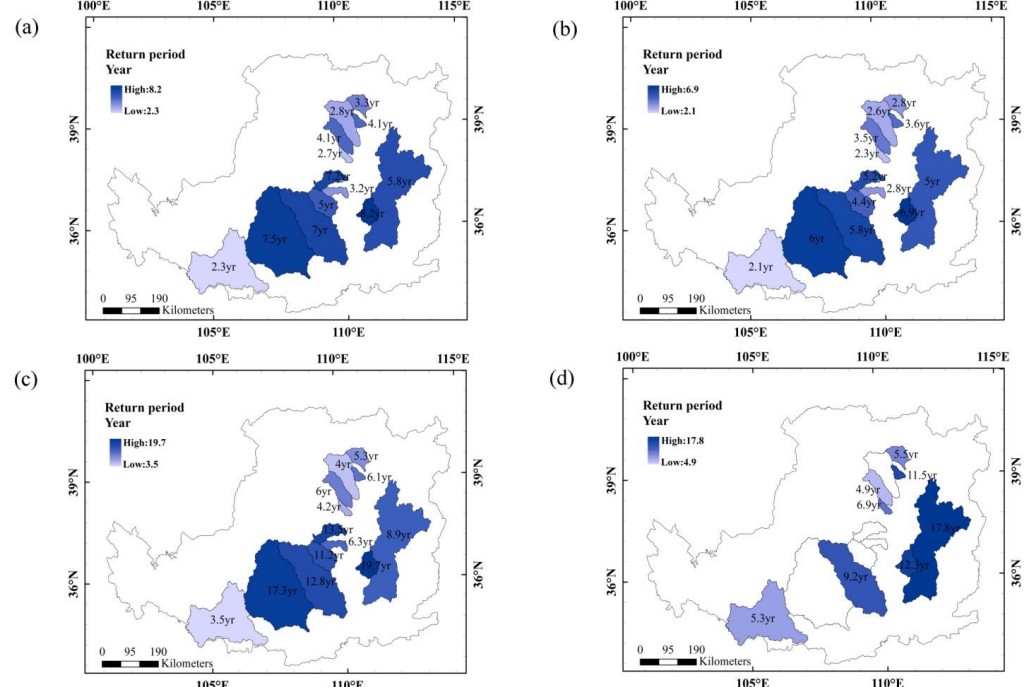

**Figure 7. Spatial distribution of the drought event return period in watersheds: (a)return period spatial distribution of dry events in 1968–1974; (b)return period spatial distribution of dry events in 1979–1983; (c) return period spatial distribution of dry events in 1991–1999; (d) return period of drought events corresponding to watersheds with significant changes in the PRR.**





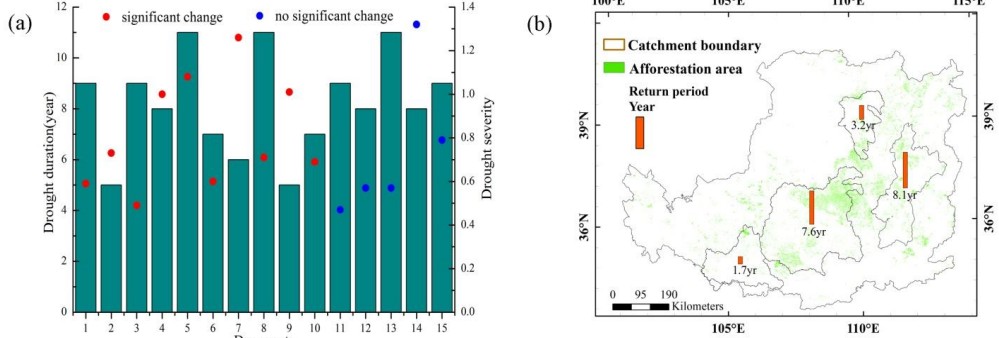

**Figure 8. (a) Characteristic of drought events with significant changes and (b) spatial distribution of the joint return period(four regions).**




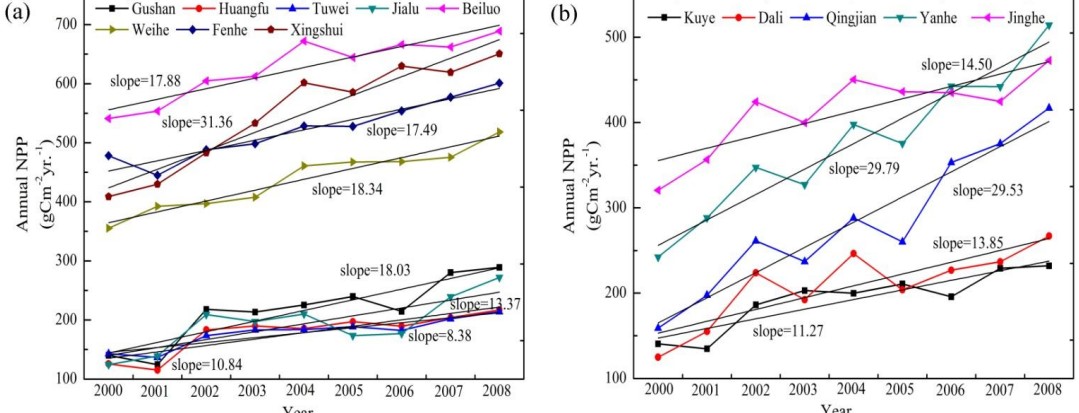

**Figure 9. Dynamic course of annual NPP in each watershed during 2000–2008: (a) interannual change trend corresponding to watersheds with significant in the PRR; (b) interannual change trend corresponding to watersheds with no significant in the PRR.**





**Table 1. Watershed characteristics and hydrological data for the study period 1961-1999**

| ID | Watershed name | Gauging station name | Area(km$^2$) | Elevation(m) | PPT(mm yr$^{-1}$) | Runoff(mm yr$^{-1}$) |
|----|----------------|----------------------|--------------|--------------|-------------------|----------------------|
| 1 | Huangfu | Huangfu | 3230 | 1162 | 400 | 37 |
| 2 | Gushan | Gaoshiya | 1260 | 1167 | 415 | 50 |
| 3 | Kuye | Wenjiachuan | 8621 | 1263 | 407 | 60 |
| 4 | Tuwei | Gaojiachuan | 3307 | 1215 | 416 | 89 |
| 5 | Jialu | Shenjiawan | 1138 | 1117 | 440 | 48 |
| 6 | Dali | Suide | 3861 | 1202 | 485 | 37 |
| 7 | Qingjian | Yanchuan | 3600 | 1186 | 502 | 38 |
| 8 | Yanhe | Ganguyi | 5857 | 1282 | 506 | 34 |
| 9 | Beiluo | Zhuangtou | 25723 | 1283 | 504 | 35 |
| 10 | Jinghe | Zhangjiashan | 43106 | 1420 | 533 | 33 |
| 11 | Weihe | Linjiacun | 30122 | 1895 | 502 | 65 |
| 12 | Fenhe | Hejin | 38728 | 1135 | 520 | 23 |
| 13 | Xinshui | Daning | 4186 | 1217 | 498 | 29 |

**ID, Watershed identification number; PPT, Annual precipitation.**





**Table 2. Common two-dimensional copula function families**

| Family | $C(u,v)$ | Parameter range |
|---|---|---|
| Frank | $-\dfrac{1}{\theta}\ln[1+\dfrac{(e^{-\theta u}-1)(e^{-\theta v}-1)}{e^{-\theta}-1}]$ | $(-\infty,\infty)$ |
| Clayton | $\max([u^{-\alpha}+v^{-\alpha}-1]^{-1/\alpha},0)$ | $[-1,\infty)\setminus\{0\}$ |
| Gumbel | $\exp(-[(-\ln u)^{\alpha}+(-\ln v)^{\alpha}])^{1/\alpha}$ | $[1,\infty)$ |
| t-copula | $\displaystyle\int_{-\infty}^{t_k^{-1}(u)}\int_{-\infty}^{t_k^{-1}(v)}\dfrac{1}{2\pi\sqrt{1-\rho^2}}[1+\dfrac{s^2-2\rho st+t^2}{k(1-\rho^2)}]^{-(k+2)/2}d_sd_t$ | $(-\infty,\infty)$ |
| Normal | $\displaystyle\int_{-\infty}^{\phi^{-1}(u)}\int_{-\infty}^{\phi^{-1}(v)}\dfrac{1}{2\pi\sqrt{1-\rho^2}}\exp[-\dfrac{s^2-2\rho st+t^2}{2(1-\rho^2)}]d_sd_t$ | $(-\infty,\infty)$ |





**Table3. The goodness-of-fit about copula function($d^2$)**

| Watershed name | Gauging station name | Normal | t-Copula | Clayton | Frank | Gumbel |
|---|---|---|---|---|---|---|
| Huangfu | Huangfu | 0.2772 | 0.2715 | 0.2593 | **0.2488** | 0.2501 |
| Gushan | Gaoshiya | 0.1180 | 0.1186 | **0.1027** | 0.1029 | 0.1054 |
| Kuye | Wenjiachuan | 0.1383 | 0.1395 | 0.2171 | **0.1266** | 0.1305 |
| Tuwei | Gaojiachuan | 0.2270 | 0.2230 | 0.3026 | **0.2199** | 0.2239 |
| Jialuo | Shenjiawan | 0.1319 | 0.1323 | 0.1469 | **0.1267** | 0.1371 |
| Dali | Suide | 0.3778 | 0.3740 | **0.3358** | 0.3438 | 0.4625 |
| Qingjian | Yanchuan | 0.1879 | 0.1888 | 0.2565 | **0.1720** | 0.1976 |
| Yanhe | Ganguyi | 0.1986 | 0.2166 | 0.2186 | **0.1979** | 0.2059 |
| Beiluo | Zhuangtou | 0.2467 | 0.2480 | 0.2376 | **0.2264** | 0.2351 |
| Jinghe | Zhangjiashan | 0.3142 | 0.3358 | 0.3668 | **0.3093** | 0.3234 |
| Weihe | Linjiacun | 0.1784 | 0.1751 | 0.1743 | 0.1604 | **0.1594** |
| Fenhe | Hejin | 0.3766 | 0.3753 | 0.4339 | **0.3726** | 0.3827 |
| Xinshui | Daning | 0.5453 | 0.5379 | 0.5567 | **0.5267** | 0.6052 |
| All | | 0.2037 | 0.1972 | 0.2141 | **0.1891** | 0.1897 |