# Peer review of "Characterizing drought in terms of changes in the precipitation-runoff relationship: a case study of the Loess Plateau, China"

_Hydrology and Earth System Sciences, 2017_

## Referee Comment (RC1) · Anonymous Referee #1 · 22 Jun 2017

General comments: This work characterizes the drought by linking climate anomaly with the change in precipitation-runoff relationship in China's Loess Plateau, and discusses the policy implications of the study to water resource management in a water-limiting environment. The study is scientifically valid, the methods and data sources are well explained, and the results are clear and well presented, though there are some aspects need to ameliorate. Overall, I would like to support the publication of this manuscript with some comments and suggestions to be addressed.

Section 2.4.1. Parameters estimation: The paper chooses seven commonly functions as the candidate margins distribution for drought duration and severity, there are some

deficiencies in fitting margin distribution function. For example, "by comparison...", I hope the authors can provide quantitative value to determine distribution functions. "drought and severity are fitted with weibull and gamma ...", the authors need to show relevant statistical indicators.

Section 2.4.2. Only the method of Squared Euclidean Distance(SED) is used to perform the goodness-of-fit of joint distribution function, I recommend the authors can adopt more methods to evaluate the fitted copula, such as root mean square error(RMSE), the Akaike information criterion(AIC)...

The English expression in this MS is sub-standard; it needs to be improved. The authors should further review the whole paper, although I have pointed some in specific suggestions. In addition, some sentences in the paper are very long, without clear phrasing, so that the reader is sometimes left wondering what the main point of the sentence was. The authors need also notice these problems.

Specific suggestions:

Page1.L4, not all readers will know that this re-vegetation is anthropogenic, you need to explicitly state this.

Page1.L5, delete "in the area".

Page3.L11,delete "reflect".

Page3.L20, as the climate is changing over what years are these long-term averages calculated?

Page4. L21, "propose use"?

Page6.L9, states that 7 dry periods are identified yet on Fig 8(a) there are 15 events. This is confusing.

Page6.L19, "In1991—1999(p=0.000) there was a significant decrease change significantly in the PRR", expression is repeated.

[Figure]

Page8.L6, "multi_year".

Page8.L10, "Compared to"

Page10.L24, hey you are introducing a new model and a new dataset in the Discussion section. This is very non-standard the structure is all over the place.

Fig 5, Precipitation, and many other hydrological variables, have the dimensions of depth / time, and you need to include the time of integration into you units. So your X-axis should have the units of mm/year. When assessing annual trends of annual (or actual E, potential E or Epan) the units are mm/year/year, as in such a plot the X-axis is years, and the Y-axis of an annual P time-series is mm/year, so the slope (or trend) of delta_Y / delta_X has the units of mm/year/year.

Suggested references:

Mishra, A. K. and Singh, V. P.: A review of drought concepts, Journal of Hydrology, 391, 202-216, 2010.

Sun, F., Lim, W., and Farquhar, G.: A general framework for understanding the response of the water cycle to global warming over land and ocean, Hydrology and Earth System Sciences, 18, 1575, 2014.

Vicente-Serrano, S. M., Beguería, S., and López-Moreno, J. I.: A multiscalar drought index sensitive to global warming: the standardized precipitation evapotranspiration index, Journal of climate, 23, 1696-1718, 2010.

Vicente-Serrano, S. M., Gouveia, C., Camarero, J. J., Beguería, S., Trigo, R., López-Moreno, J. I., Azorín-Molina, C., Pasho, E., Lorenzo-Lacruz, J., and Revuelto, J.: Response of vegetation to drought time-scales across global land biomes, Proceedings of the National Academy of Sciences, 110, 52-57, 2013.

Vicente-Serrano, S. M., Lopez-Moreno, J.-I., Beguería, S., Lorenzo-Lacruz, J., Sanchez-Lorenzo, A., García-Ruiz, J. M., Azorin-Molina, C., Morán-Tejeda, E., Revuelto, J., and Trigo, R.: Evidence of increasing drought severity caused by tempera-
ture rise in southern Europe, Environmental Research Letters, 9, 044001, 2014.

---

## Referee Comment (RC2) · Anonymous Referee #2 · 2 Jul 2017

General comments: The authors analyze the drought impacts on the runoff ratio in China's Loess Plateau. The climate anomaly, relationships between precipitation-runoff, the implications for ecosystem, and the water resource management were discussed in the manuscript. The structure of the manuscript and the problems description are well organized, but there are several serious flaws in the data analysis, methods description, and interpretations of results. Thus, this version of the manuscript can not be accepted for publication in HESS.

First of all, the amount of the water consumption for the local communities (domestic and industrial usage) is vital for the runoff ratio in the study period, especially for during

the drought. The authors should at least investigate the changes in the water supply for the local communities.

The precipitation -runoff relationships can be influenced by the land use, surface water diversion, irrigation scheme, groundwater abstraction, and the water storage in the (sub) catchment. These issues should be addressed for identifying the influence of drought on the water yield.

Section 2.2 The proposed classification method of drought events, drought periods, the interpretations of results, and the upscale processes from 13 sub catchments to regional precipitation anomaly are not clear enough to support the publication of this version of the manuscript in HESS.

The NPP estimation based on the remote sensing data (2000-2008) could not support the analysis results of the drought on the ecosystem from 1961 to 1999. The authors need to find at least the data in one of the main drought period defined in this manuscript and another normal period to illustrate the difference for determining the drought impacts.

The English should be substantial improved to a certain level that the readers can not misunderstand the correct information.

Specific comments: Affiliation: Shaanxi? should be Shanxi.

Page 1, line 1, "is" should be "are". Page 1, line 5, only the re-vegetation that makes the drought a major concern? Page 1, line 12 delete the "around" after "(precipitation" Page 1, line 13-14,"NPP" and "PRR" should not be abbreviation in first appearance. page 2, line 9-11, weird sentence. Page 2, line 30, replace the "with" with "by". Page 3, line 25, please indicate the data length or periods. Page 3, line 27, website in the bracket does not match the text. Page 4, line 2, replace "its" with "in". Page 4, line 4, conditions 2 should be page 6, clarified. Page 6, line 21, please indentify the time period for "long term". Page 7, line 25-27, long sentence. Section 3.3, please re-write

[Figure]

the first paragraph. Page 8, line 10, where are the basins with significant changes in precipitation in table 1? Page 9, line 5, replace the "as well as " with "and " Page 9, line 11, should be "http://www.mwr.gov.cn"

Figure 1, where is the Yellow river? it is indicated on the up-left small plot that the Yellow river flows through the loess plateau. Figure 2, Do you use the average of rainfall for the 13 watersheds? The description of drought events for condition 1 and 2 in section 2.2 may not be applied on the year 1974, when the 3- year moving average should be lowest in the first main drought period. But the 3-year moving average in 1970 in the figure is lowest. Figure 3, What are the historical records? Apparently, the historical records in three plots are different, why? better to use the same scale for x-axis in three plots. Figure 5, the drought periods in different sub-catchments are not identical, why? again, what are these historical records? Figure 7, what is the drought event corresponding to the return period in figure7d? Figure 8, at least show the whole legend of the figure 8a. is it the average return period of the drought events in 8b? Figure 9, add "change" after the significant in the caption.

---

## Author Comment (AC1) · 5 Aug 2017

**To Reviewer #1:**

**General Comments:**

*It has been a pleasure reading through this contributions. This work characterizes the drought by linking climate anomaly with the change in precipitation-runoff relationship in China's Loess Plateau, and discusses the policy implications of the study to water resource management in a water-limiting environment. The study is scientifically valid, the methods and data sources are well explained, and the results are clear and well presented, though there are some aspects need to ameliorate. Overall, I would recommend this manuscript for publication in Hydrology and Earth System Sciences, with some comments and suggestions.*

**[Response]** We thank the reviewer for supporting the publication of this MS. The MS will be revised carefully after the reviewer's comments and suggestions, with the detailed responses as followed.

**[Reviewer #1 Comment 1]** *Section 2.4.1. Parameters estimation: The paper chooses seven commonly functions as the candidate margins distribution for drought duration and severity, there are some deficiencies in fitting margin distribution function. For example, "by comparison…", I hope the authors can provide quantitative value to determine distribution functions. "drought and severity are fitted with weibull and gamma …", the authors need to show relevant statistical indicators.*

**[Response]** According to the comment, we will provide quantitative value to determine the marginal distribution function. We will use Root Mean Square Error (RMSE) and Kolmogorov-Smirnov (K-S) test to select the best fitted distribution. Table R1.1 lists the estimated parameters and the results of goodness of fit test. We can find that not all the distributions pass the K-S test at the 95% ($\alpha$=0.05) significant level. Further considering RMSE, the best fitted marginal distributions of duration, severity are weibull and gamma, respectively, which are marked with bold fonts and underlined in Table1 R1.1.

Table R1.1 Parameters and goodness of fit of the marginal distributions

| Distribution | | Parameters | RMSE | K-S test | |
| --- | --- | --- | --- | --- | --- |
| | | | | Statistic | $p$_value |
| **(Duration)** | Exponential(exp) | param_exp =3.714 | 0.617 | 0.223 | 0.306 |
| | Gamma(gam) | param_gam1=1.421 param_gam2=2.614 | 0.625 | 1 | 0 |
| | Log-normal(lno) | param_lno1=3.714 param_lno2=3.253 | 0.668 | 1 | 0 |
| | Extreme value(ev) | param_ev1=5.319 param_ev2=3.089 | 0.498 | 1 | 0 |
| Generalized extreme value(gev) | | param_gev1=3.722 param_gev2=0.008 param_gev3=1.002 | 0.668 | 1 | 0 |
| | Poisson(poission) | param_poission =3.714 | 0.645 | 1 | 0 |
| | **Weibull(wbl)** | **param_wbl1=3.975 param_wbl2=1.213** | **0.581** | **0.248** | **0.231** |
| **(Severity)** | Exponential(exp) | param_exp =0.309 | 0.112 | 0.280 | 0.883 |
| | **Gamma(gam)** | **param_gam1=3.690 param_gam2=0.084** | **0.090** | **0.267** | **0.892** |
| | Log-normal(lno) | param_lno1=0.310 param_lno2=0.254 | 0.237 | 0.423 | 0.423 |
| | Extreme value(ev) | param_ev1=0.393 param_ev2=0.168 | 0.127 | 0.280 | 0.883 |
| Generalized extreme value(gev) | | param_gev1=0.111 param_gev2=0.119 param_gev3=0.227 | 0.092 | 0.276 | 0.885 |
| | Poisson(poission) | param_poission =0.310 | 0.329 | 0.714 | 0.028 |
| | Weibull(wbl) | param_wbl1=0.351 param_wbl2=2.071 | 0.098 | 0.286 | 0.822 |

**[Reviewer #1 Comment 2]** *Section 2.4.2. Only the method of Squared Euclidean Distance(SED) is used to perform the goodness-of-fit of joint distribution function, I recommend the authors can adopt more methods to evaluate the fitted copula, such as root mean square error(RMSE), the Akaike information criterion(AIC)...*

**[Response]** We thank the reviewer for this comment. Besides the method of Squared Euclidean Distance (SED), we will adopt root mean square error (RMSE) and the Akaike information criterion (AIC) to further evaluate the fitted copula. As shown in Table R1.2, Frank-copula is the optimal joint distribution function in most watersheds of this study except for Jialu, Dali and Beiluo watersheds. The optimal goodness of fit

of different methods are also marked with bold fonts and underlined.

Table R1.2 The goodness-of-fit about copula function

| ID | Normal | | | t-Copula | | | Clayton | | | Frank | | | Gumbel | | |
|---|---|---|---|---|---|---|---|---|---|---|---|---|---|---|---|
| | $d^2$ | AIC | RMSE | $d^2$ | AIC | RMSE | $d^2$ | AIC | RMSE | $d^2$ | AIC | RMSE | $d^2$ | AIC | RMSE |
| 1 | 0.277 | 1.481 | 0.233 | 0.272 | 1.088 | 0.224 | 0.259 | 1.251 | 0.228 | **0.249** | **1.044** | **0.223** | 0.250 | 1.585 | 0.235 |
| 2 | 0.118 | -3.912 | 0.154 | 0.119 | -3.294 | 0.143 | **0.103** | -2.961 | 0.154 | **0.103** | **-3.928** | **0.143** | 0.105 | -3.753 | 0.145 |
| 3 | 0.138 | -1.868 | 0.166 | 0.140 | -1.819 | 0.167 | 0.217 | 0.836 | 0.208 | **0.127** | **-2.403** | **0.159** | 0.131 | -2.220 | 0.162 |
| 4 | 0.227 | 1.103 | 0.213 | 0.223 | 0.997 | 0.211 | 0.303 | 2.828 | 0.246 | **0.220** | **0.912** | **0.210** | 0.224 | 1.022 | 0.211 |
| 5 | 0.132 | -2.129 | 0.169 | 0.132 | -2.112 | 0.163 | 0.147 | -1.590 | 0.171 | **0.127** | -1.936 | 0.162 | 0.137 | **-2.328** | **0.159** |
| 6 | 0.378 | 5.186 | 0.275 | 0.374 | 5.116 | 0.274 | **0.336** | 4.525 | 0.259 | 0.344 | **4.436** | **0.262** | 0.462 | 6.062 | 0.304 |
| 7 | 0.188 | -0.360 | 0.194 | 0.189 | -0.335 | 0.194 | 0.257 | 1.197 | 0.227 | **0.172** | **-0.108** | **0.186** | 0.198 | -0.800 | 0.199 |
| 8 | 0.199 | 0.685 | **0.199** | 0.217 | 1.291 | 0.208 | 0.219 | 1.355 | 0.209 | **0.198** | 0.661 | **0.199** | 0.206 | 0.936 | 0.202 |
| 9 | 0.247 | 2.802 | 0.222 | 0.248 | 2.845 | 0.223 | 0.238 | 2.503 | 0.218 | **0.226** | 2.417 | 0.217 | 0.235 | **2.117** | **0.213** |
| 10 | 0.314 | 5.580 | 0.259 | 0.336 | 6.180 | 0.249 | 0.367 | 6.974 | 0.271 | **0.309** | **5.438** | **0.251** | 0.323 | 5.834 | 0.254 |
| 11 | 0.178 | -0.619 | 0.179 | 0.175 | -1.15 | 0.189 | 0.174 | -0.711 | 0.187 | **0.160** | **-1.182** | **0.176** | 0.159 | -0.735 | 0.187 |
| 12 | 0.377 | 5.164 | **0.274** | 0.375 | 5.140 | 0.275 | 0.434 | 6.156 | 0.295 | **0.373** | **5.089** | **0.274** | 0.383 | 5.278 | 0.277 |
| 13 | 0.545 | 10.541 | 0.330 | 0.538 | 10.419 | 0.328 | 0.557 | 10.727 | 0.334 | **0.527** | **10.229** | **0.325** | 0.605 | 11.480 | 0.348 |
| All | 0.204 | 4.974 | 0.265 | 0.197 | 5.642 | 0.288 | 0.214 | 5.113 | 0.301 | **0.189** | **4.312** | **0.249** | 0.190 | 5.230 | 0.254 |

**[Reviewer #1 Comment 3]***The English expression in this MS is sub-standard; it needs to be improved. The authors should further review the whole paper, although I have pointed some in specific suggestions. In addition, some sentences in the paper are very long, without clear phrasing, so that the reader is sometimes left wondering what the main point of the sentence was. The authors need also notice these problems.*

**[Response]** We will ask a native English speaking scientist to help us with the language of the revised MS.

**Specific suggestions:**

**[Reviewer #1 Suggestion 1]** *Page1.L4, not all readers will know that this re-vegetation is anthropogenic, you need to explicitly state this.*

**[Response]** Accordingly, we will explain the details of re-vegetation. China experienced serve droughts in 1997 and serious floods in 1998, causing serious economic and environmental damage (Tian et al., 2016). In the wake of these disasters, the Chinese government took unprecedented conservation measures (Xu and Cao, 2001), one of which was the Grain for Green Program (GGP) introduced in 1999 to protect the degraded environment (Zhang et al., 1999). The objective of this program was to convert cropland to plantations and grassland on steep slopes by compensating farmers with subsidies.

**[Reviewer #1 Suggestion 2]** *Page1.L5, delete "in the area".*

**[Response]** We will delete "in the area" accordingly. The sentence will be changed to "*This case study characterizes drought by linking climate anomaly with the change in precipitation-runoff relationships, in the Loess Plateau of China, a water-limited region where re-vegetation makes drought a major concern.*"

**[Reviewer #1 Suggestion 3]** *Page3.L11,delete "reflect".*

**[Response]** We will delete "reflect" accordingly. The sentence will be changed to "*So analyzing drought characteristics based on the response of the precipitation-runoff relationship (PRR) change to multi-year dry periods is of great importance in estimating the effect of drought and the ecological re-construction of the whole Loess Plateau.*"

**[Reviewer #1 Suggestion 4]** *Page3.L20, as the climate is changing over what years are these long-term averages calculated?*

**[Response]** We will clearly in the revised MS that the long-tem averages is for the period of 1960–2000.

**[Reviewer #1 Suggestion 5]** *Page4. L21, "propose use"?*

**[Response]** Accordingly, here we will modify the sentence to "we use the copula function (Shiau, 2006)."

**[Reviewer #1 Suggestion 6]** *Page6.L9, states that 7 dry periods are identified yet on Fig 8(a) there are 15 events. This is confusing.*

**[Response]** The identified 7 dry periods in Page 6 L9 are for the whole Loess Plateau. However, 15 events in Fig8(a) are for 13 watersheds. We will clarify in the revised MS as: "*Based on the drought identification method developed in this study, 7 dry periods are identified (including the main dry period and single dry period) on the whole Loess Plateau during 1961-1999. The purpose of this study is to focus on the change of the PRR in the main dry period. So further considering the variability of the PRR during dry period in each watershed (section 3.3), there are 15 dry periods (including significant changes and non-significant in the ) in 13 watersheds with drought regressions fell under the total regression lines.*"

**[Reviewer #1 Suggestion 7]** *Page6.L19, "In1991ăĂˇT1999(p=0.000) there was a significant decrease change significantly in the PRR", expression is repeated.*

**[Response]** Accordingly, we will modify the sentence to "In 1991–1999 (p=0.000) there was a significant decrease change in the PRR."

**[Reviewer #1 Suggestion 8]** *Page8.L6, "multi_year".*

**[Response]** Accordingly, we will revise "multi_yeat" to "multiyear".

**[Reviewer #1 Suggestion 9]** *Page8.L10, "Compared to"*

**[Response]** Accordingly, we will revise this sentence as "*Compared with the annual average precipitation (Table1) in other basins where significant changes occurred, these watersheds where there were no significant changes in precipitation-runoff relationship (Kuye, Dali, Qingjian, Yanhe, Jinghe) had higher precipitation.*"

**[Reviewer #1 Suggestion 10]** *Page10.L24, hey you are introducing a new model and a new dataset in the Discussion section. This is very non-standard the structure is all over the place.*

**[Response]** We agree with the reviewer that, in standard, the Net Primary Production (NPP) data that derived with terrestrial Carnegie-Ames-Stanford Approach (CASA) we used in the discussion section in Page 10 L24 should be first explained in section 2.1.

However, we will replace the NPP in this section with satellite-derived Leaf Area Index (LAI) following the comment of Reviewer #2 (See detail in the response to Reviewer #2 general comment 4). In line with this comment, we will explain the LAI data in section 2.1 in the revised MS.

**[Reviewer #1 Suggestion 11]** *Fig 5, Precipitation, and many other hydrological variables, have the dimensions of depth / time, and you need to include the time of integration into you units. So your X-axis should have the units of mm/year. When assessing annual trends of annual (or actual E, potential E or Epan) the units are mm/year/year, as in such a plot the X-axis is years, and the Y-axis of an annual P time-series is mm/year, so the slope (or trend) of delta_Y / delta_X has the units of mm/year/year.*

**[Response]** Accordingly, we will revise the X-axis, Y-axis to P (mm/year) and Runoff (mm/year) respectively in revised MS as Figure R1.1.

[Figure]

Figure R1.1. The annual precipitation-runoff scatter plot in each watershed.

[Figure]

Figure R1.1. (continued).

[Figure]

Figure R1.1. (continued).

**References**

Tian, F., Feng, X., Zhang, L., Fu, B., Wang, S., Lv, Y., and Wang, P.: Effects of revegetation on soil moisture under different precipitation gradients in the Loess Plateau, China, Hydrology Research, 2016. nh2016022, 2016.

Xu, J. and Cao, Y.: The socioeconomic impacts and sustainability of the SLCP, Implementing the Natural Forest Protection Program and the Sloping Land Conversion Program: Lessons and Policy Recommendations. CCICED-Task Force on Forests and Grasslands. Beijing: China Forestry Publishing House, 2001.

Zhang, X. P., Zhang, L., McVicar, T. R., Van Niel, T. G., Li, L. T., Li, R., Yang, Q., and Wei, L.: Modelling the impact of afforestation on average annual streamflow in the Loess Plateau, China, Hydrological Processes, 22, 1996-2004, 2008.

---

## Author Comment (AC2) · 5 Aug 2017

**To Reviewer #2:**

**General Comments:**

*The authors analyze the drought impacts on the runoff ratio in China's Loess Plateau. The climate anomaly, relationships between precipitation-runoff, the implications for ecosystem, and the water resource management were discussed in the manuscript. The structure of the manuscript and the problems description are well organized, but there are several serious flaws in the data analysis, methods description, and interpretations of results. Thus, this version of the manuscript can not be accepted for publication in HESS.*

**[Response]** We thank the reviewer for the comments. We have carefully considered all the reviewer's comments as followed. We believe the MS will be highly improved after the revision, with the flaws raised by the reviewer all being addressed.

**[Reviewer #2 Comment 1]** *First of all, the amount of the water consumption for the local communities (domestic and industrial usage) is vital for the runoff ratio in the study period, especially for during the drought. The authors should at least investigate the changes in the water supply for the local communities.*

**[Response]** We agree that the amount of the water consumption for the local communities is vital for the runoff ratio, especially for the drought period. For example, Bouwer et al. (2006) concluded that increasing water consumption for irrigation and hydropower caused runoff variability is three times higher than variation in runoff under climate change in which the area is densely populated and the main agricultural irrigation area in India . However, the water consumption for the local communities is not a major issue in our study area, which is composed by 13 loess hilly catchments.

With the Mu Us Desert in the northwest and the WeiHe Plain in the southeast, the catchments in our study lie in the most hilly part of the Loess Plateau. Water consumption for the local communities is for the residential area, locating mainly at the flat area at the outlet of the catchments (See the distribution of resident area in

Figure R2.1). Moreover, the population tend to move from the catchment area to the main big cities, locating in the mainstream of the River Basin in the WeiHe Plain (i.e. Baoji, Xi'an, 57.35% ) because of the accelerated urbanization process in this area since the 1980s (Hu et al., 2001).

In line with our consideration, in the studies of runoff variability of the same catchment, water supply for the local communities was also not included as factor influencing runoff. Instead, anthropogenic factors of the change in runoff 13 catchments   such as terrace building, soil conservation measures and so on (Wang et al., 2009; Shi et al., 2013; Chang et al., 2015). We will add the explanations in the revised MS.

[Figure]

Figure R2.1. Study area

**[Reviewer #2 Comment 2]** *The precipitation-runoff relationships can be influenced by the land use, surface water diversion, irrigation scheme, groundwater abstraction, and the water storage in the (sub) catchment. These issues should be addressed for identifying the influence of drought on the water yield.*

**[Response]** We agree with the reviewer that the precipitation-runoff relationships can be influenced by factors other than climate conditions. We have carefully considered each possible factor in our study as followed.

The catchments lie in the most hilly part of the Loess Plateau, vegetation in the catchments is mostly rain-fed, thus irrigation scheme in the study area can be neglected (Feng et al., 2016). Loess thickness of the catchments area is more than 100m (Derbyshire et al., 1998), the groundwater is less impacted by the surface eco-hydrological process, thus groundwater recharge and groundwater discharge are not considered in the study area. Finally, the possible surface water diversion and water storage is in the residential area at the outlet these catchments (See detailed explanation in the response of the Reviewer#2 Comment 1), therefore is not included as the impact factor of the precipitation-runoff relationships.

However, soil conservation measures, including terraces construction and sediment-trap dams building, have been implemented in the Loess Plateau since the 1950s (Wang et al., 2016). We will add the description on the influence of these human-stimulated measurements to precipitation-runoff relationships in the revised MS. The partial correlation method will be used to isolate the impact of human-stimulated measurements from climatic factor.

For the entire period of 1982–1999, the runoff had a decreasing trend (Figure R2.2).Terrace construction played an important role in the runoff ratio reduction from 1980s to 1990s ($p$=0.048, Figure R2.3). Other anthropogenic factors including dam construction, tree and pasture plantation had not caused the change in runoff ratio in this period. Terra construction contributed 25% of the runoff ratio reduction in 1990s, so the drought is the major factor of the runoff reduction in the study area. We will add the analysis in the revised MS.

[Figure]

Figure R2.2. Trend of annual runoff ratio during 1982−1999

[Figure]

Figure R2.3. Anthropogenic factors of runoff ratio change in 1980s vs. 1990s (ΔTerra, ΔDam, ΔTree and ΔPasture are changes of percentage area for terraces, check-dams, tree planting and natural pasture, respectively. ΔLAI is change of GIMMS LAI for each catchment)

**[Reviewer #2 Comment 3]***Section 2.2 The proposed classification method of drought events, drought periods, the interpretations of results, and the upscale processes from 13 sub catchments to regional precipitation anomaly are not clear enough to support the publication of this version of the manuscript in HESS.*

**[Response]** We thank the reviewer for this comment. We will rewrite section 2.2 to clarify the classification method, and it will appear as followed in the revised MS.

*"In this study, we defined drought based on annual precipitation for two aspects. On the one hand, the amount of precipitation is the most important climatic element necessary as an input into drought (Mishra and Singh, 2010). What's more, we are interested in testing whether the runoff response differs for multiyear droughts, therefore the runoff should not be considered to identify drought.*

*We first calculated the precipitation anomaly (PA) in the whole Loess Plateau its separate watersheds, the anomaly series were divided by the mean annual precipitation and smoothed with a 3-year moving average. Positive PAs indicate that the observed precipitation is greater than the median, while the negative PAs is below the median, which implied the drought may occur under this circumstance. Each drought event is characterized by two main properties: drought duration and drought severity. Studies have shown that the drought events with shorter duration but stronger intensity or lower intensity but longer duration would also cause serious water-supply and other drought-related problems (Shiau, 2006; Naresh et al., 2009). Therefore, the basic rules for identifying drought events in this study are (1) A single year's PA $\leqslant$ -10% or (2) Mean PAs of more than three consecutive years less than 0, note the start year's PA of the drought period remained a negative value.*

*Cumulated PAs during the drought period is used to measure the drought severity in this study (for convenience, drought severity is multiplied by -1 to obtain a positive value). Based on the rules above-mentioned, we identified all dry events in each watershed. In order to reflect the response of PRR to drought events over the years, we must ensure that the dry periods are sufficiently long and severe, in the subsequent analysis, we only consider the drought events with drought duration $\geqslant$ 5years and*

*mean annual PA ≤ -5% during the drought period. Finally, the dry events will be classified two types: the main dry period and the single dry period.*

*We used the Kolmogorov-Smirnov (K-S) (Massey, 1951) test to determine whether annual precipitation and runoff data followed an roughly normal distribution, if they did not, they were transformed with a Box-Cox transformation (Box and Cox,1964). After identifying the main drought events, we tested whether the PRR change was statistically significant compared to the historic record using Student's T-test."*

**[Reviewer #2 Comment 4]** *The NPP estimation based on the remote sensing data (2000-2008) could not support the analysis results of the drought on the ecosystem from 1961 to 1999. The authors need to find at least the data in one of the main drought period defined in this manuscript and another normal period to illustrate the difference for determining the drought impacts.*

**[Response]** We agree with the reviewer that we should find the data that differs between the drought period and the normal period to illustrate the drought impacts. Due to the lack of NPP data before 1999, we will use AVHRR GIMMS LAI3g for the comparison covering long term of 1982−1999 in the revised MS. We choose the drought period of 1991−1999 as an example, we find that LAI significantly ($p$=0.032, Student's T-test) decrease in 1991−1999 compared to that of 1984−1990. We will include this new analysis in the revised MS.

[Figure]

Figure R2.4. Trend of LAI during 1982−1999

**[Reviewer #2 Comment 5]** *The English should be substantial improved to a certain level that the readers can not misunderstand the correct information.*

**[Response]** We will ask a native English speaking scientist to help us with the language in revised MS.

**Specific comments:**

**[Reviewer #2 Comment 1]** *Affiliation: Shaanxi? should be Shanxi.*

**[Response]** Affiliation in this manuscript is Shaanxi. Shanxi is a different province in China , which is not related to the MS.

**[Reviewer #2 Comment 2]** *Page 1, line 1, "is" should be "are".*

**[Response]** Accordingly, the sentence will be changed to "*The frequency and intensity of drought are increasing dramatically with global warming.*"

**[Reviewer #2 Comment 3]** *Page 1, line 5, only the re-vegetation that makes the drought a major concern?*

**[Response]** The vegetation restoration programme in China is implemented as the biggest investment to restore the ecosystem in the developing country. The sustainability of vegetation restoration in the Loess Plateau due to the limitation of water is a major concern of scientific research and policy maker in this area (Feng et al., 2016). We will clarify this point in the revised MS.

**[Reviewer #2 Comment 4]** *Page 1, line 12 delete the "around" after "precipitation"*

**[Response]** We will delete "around" after "precipitation" accordingly.

**[Reviewer #2 Comment 5]** *Page 1, line 13-14,"NPP" and "PRR" should not be abbreviation in first appearance.*

**[Response]** Accordingly, we will change the sentence to "*At the same time, the growth ratio of annual Leaf Area Index (LAI) is also susceptible to prolonged drought, the growth ratio is lower in these watersheds which had a significant change in*

*precipitation-runoff relationship (PRR)*".   NPP will be replaced with LAI, which the detailed explanation being found in the response to Reviewer #2 general comment 4.

**[Reviewer #2 Comment 6]** *page 2, line 9-11, weird sentence.*

**[Response]** We are sorry for the confused expression. The sentence will be changed to "*Soil moisture indicator (Xia et al., 2014), crop drought indicator (Duff et al.,1997) and the crop water demand indicators are used to identify agricultural drought, which is a period with dry soils that results from below-average precipitation, intense but less frequent rain events, or above-normal evaporation and all of these would lead to reduced crop production and plant growth.*"

**[Reviewer #2 Comment 7]** *Page 2, line 30, replace the "with" with "by".*

**[Response]** Accordingly, we will revise the sentence as "*so the shift in PPR caused with an extended drought will eventually have an adverse effect on the ecosystem service of water yield.*"

**[Reviewer #2 Comment 8]** *Page 3, line 25, please indicate the data length or periods.*

**[Response]** Accordingly, we will clarify in the revised MS that the data length is from 1961 to 1999.

**[Reviewer #2 Comment 9]** *Page 3, line 27, website in the bracket does not match the text.*

**[Response]** Following the comment, we will revise the website in the bracket to http://www.yellowriver.gov.cn/.

**[Reviewer #2 Comment 10]** *Page 4, line 2, replace "its" with "in".*

**[Response]** Following this comment, we will modify the sentence to "*we first calculated the precipitation anomalies in the whole Loess Plateau in separate watersheds, the anomaly series were divided by the mean annual precipitation and smoothed with a 3-year moving average.*"

**[Reviewer #2 Comment 11]** *Page 4, line 4, conditions 2 should be page 6, clarified.*

**[Response]** Accordingly, we will check the condition 2 in Page 4, line 4 in the MS. As one of the basic rules for identifying drought events in this study, we think put it in section 2.2 (Page 4, line 4) is more appropriate.

**[Reviewer #2 Comment 12]** *Page 6, line 21, please indentify the time period for "long term".*

**[Response]** Accordingly, we will indentify in the revised MS that the "long term" is for the period of 1961−1990.

**[Reviewer #2 Comment 13]** *Page 7, line 25-27, long sentence.*

**[Response]** Accordingly, the sentence in Page 7, line 25-27 will be revised to "D*uring drought period in 1968−1974, the return period was about 7.29 years under the corresponding drought characteristics. So the next drought events similar to drought period in 1968−1974 is happened around in 1980. In 1979−1983, the drought severity reached 0.41, which was close to the predicted return period.*"

**[Reviewer #2 Comment 14]** *Section 3.3, please re-write the first paragraph.*

**[Response]** We will re-write the first paragraph in section 3.3 as followed:

"*Fig.5 demonstrates the range of changes in the PRR under sustained precipitation reduction. According to the direction of change, the dry period regression line is mainly located upward or downward the overall regression, the change of the PRR in the studied 13 watersheds show no significant when the regression line of the dry period was above the total regression line. In the 15 times dry events under the regression line in the case of the total regression, there are 9 times significant change in the PRR (p < 0.05), accounting for about 60% of the total situation. In this case, the dry period regression line lies lower than nearly all the other points indicating unprecedentedly low runoff generation rates for the given rainfall. Thus in a sequence of years with decreased precipitation, we can conclude that lower runoff not only relate to the lower precipitation, but also less runoff than expected caused by the*

*multiyear drought period.*"

**[Reviewer #2 Comment 15]** *Page 8, line 10, where are the basins with significant changes in precipitation in table 1?*

**[Response]** We apologized for the confusion. "significant changes" in Page 8 line 10 refers to those watersheds which have a significant change in the PRR. Comparing the annual mean precipitation in separate watersheds during 1961−1999 (Figure R2.5), we can find that multiyear drought is more likely to cause a significant change in the PRR in the basins with less precipitation. We will clarify this point in the revised MS.

[Figure]

Figure R2.5. Annual precipitation in each catchment

**[Reviewer #2 Comment 16]** *Page 9, line 5, replace the "as well as " with "and "*

**[Response]** Accordingly, we will modify the sentence to "*prolonged multi-year drought has caused significant damages both in the natural environments and development of the human societies (Belal et al., 2014).*"

**[Reviewer #2 Comment 17]** *Page 9, line 11, should be "http://www.mwr.gov.cn"*

**[Response]** Accordingly, we will revise the website to "http://www.mwr.gov.cn".

**[Reviewer #2 Comment 18]** *Figure 1, where is the Yellow river? it is indicated on the up-left small plot that the Yellow river flows through the loess plateau.*

**[Response]** Yellow River flows through the Loess Plateau. We will add the note of Yellow River on the up-left small plot (Figure R2.6).

[Figure]

Figure R2.6. The study are

**[Reviewer #2 Comment 19]** *Figure 2, Do you use the average of rainfall for the 13 watersheds? The description of drought events for condition 1 and 2 in section 2.2 may not be applied on the year 1974, when the 3- year moving average should be lowest in the first main drought period. But the 3-year moving average in 1970 in the figure is lowest.*

**[Response]** Yes, we used the average of rainfall for the 13 watersheds in Figure 2. After re-examining the calculation of the original data, we found we made a mistake in computing the 3-year moving average. After revising the results, the first main drought period is defined in 1970–1974. As shown in Figure R2.7, the rainfall anomaly in 1974 is -23%, which is accord with the condition 1 in section 2.2. In the first main drought period, the 3- year moving average of 1974 is only smaller than in 1972. We will correct the mistake in the revised MS.

[Figure]

Figure R2.7. average annual precipitation in every catchment

**[Reviewer #2 Comment 20]** *Figure 3, What are the historical records? Apparently, the historical records in three plots are different, why? better to use the same scale for x-axis in three plots.*

**[Response]** Historical records in Figure 3 refer to the annual precipitation-runoff scatter plot in a period of 1961−1999 except for a certain main drought period. For example, when the drought occurred in 1970−1974, the corresponding historical record includes precipitation-runoff scatter plot in 1961−1969 and 1975−1999. When the drought occurred in 1991−1999, the corresponding historical record refers to precipitation-runoff scatter plot in 1961−1990. Due to there are three different multiyear dry period in the Loess Plateau during 1961−1999, the corresponding historical records are different in three plots. We will clarify this point in the revised MS and use the same scale for x-axis in three plots in the revised MS as Figure R2.8.

[Figure]

Figure R2.8. Precipitation-runoff relationships during drought periods: (a, b) no significant change in precipitation-runoff relationship and (c) significant downward change in precipitation-runoff relationship.

**[Reviewer #2 Comment 21]** *Figure 5, the drought periods in different sub-catchments are not identical, why? again, what are these historical records?*

**[Response]** The drought period is spatial varied, as Figure 5 showed. The drought period of each catchment is identified with the local data of precipitation and change of precipitation-runoff relationship.

Similar with Figure 3, the historical records refer to the annual precipitation-runoff scatter plot in a period of 1961−1999 except for a certain main drought period, which the detailed explanation being found in the response to Reviewer #2 comment 20.

**[Reviewer #2 Comment 22]** *Figure 7, what is the drought event corresponding to the return period in figure7d?*

**[Response]** The drought event corresponding to the return period in figure 7d is drought duration and severity that caused a significant change in the PRR in 8 watersheds. We will clarify this point in the revised MS.

**[Reviewer #2 Comment 23]** *Figure 8, at least show the whole legend of the figure 8a. is it the average return period of the drought events in 8b?*

**[Response]** Following this comment, we will show the whole legend of the figure 8a in the revised manuscript. It is the average return period of the drought events in 8b. We will clarify in the revised MS as "*Based on the spatial distribution of drought events in the Loess plateau, the 13 watersheds were further divided into four regions. Similarly, we calculated the return period in these four regions according to their main dry periods during 1961–1999, and showed the average return period of the drought events in 8b.*"

**[Reviewer #2 Comment 24]** *Figure 9, add "change" after the significant in the caption.*

**[Response]** Accordingly, we will add "change" after the significant in the caption of Figure 9.

**References**

Bouwer, L., Aerts, J., Droogers, P., and Dolman, A.: Detecting the long-term impacts from climate variability and increasing water consumption on runoff in the Krishna river basin (India), Hydrology and Earth System Sciences Discussions, 3, 1249-1280, 2006.

Chang, J., Wang, Y., Istanbulluoglu, E., Bai, T., Huang, Q., Yang, D., and Huang, S.: Impact of climate change and human activities on runoff in the Weihe River Basin, China, Quaternary International, 380, 169-179, 2015.

Feng, X., Fu, B., Piao, S., Wang, S., Ciais, P., Zeng, Z., Lü, Y., Zeng, Y., Li, Y., and Jiang, X.: Revegetation in China's Loess Plateau is approaching sustainable water resource limits, Nature Climate Change, 6, 1019-1022, 2016.

Hu, S., Zhi-mao, G., and Jun-ping, Y.: The impacts of urbanization on soil erosion in the Loess Plateau region, Journal of Geographical Sciences, 11, 282-290, 2001.

Shi, C., Zhou, Y., Fan, X., and Shao, W.: A study on the annual runoff change and its relationship with water and soil conservation practices and climate change in the

middle Yellow River basin, Catena, 100, 31-41, 2013.

Wang, X.-j., Cai, H.-j., Zhang, X., Wang, J., and Zhai, J.: Analysis of changing characteristics and tendency of runoff and sediment transport in Huangfuchuan River watershed, Res Soil Water Conserv, 16, 222-226, 2009.

Derbyshire, E., Meng, X., and Kemp, R. A.: Provenance, transport and characteristics of modern aeolian dust in western Gansu Province, China, and interpretation of the Quaternary loess record, Journal of arid environments, 39, 497-516, 1998.

Mishra, A. K. and Singh, V. P.: A review of drought concepts, Journal of hydrology, 391, 202-216, 2010.

Naresh Kumar, M., Murthy, C., Sesha Sai, M., and Roy, P.: On the use of Standardized Precipitation Index (SPI) for drought intensity assessment, Meteorological applications, 16, 381-389, 2009.

Shiau, J.: Fitting drought duration and severity with two-dimensional copulas, Water resources management, 20, 795-815, 2006.

Searcy, J. K., Hardisoni, C.H., 1960. Double mass curves, Geological Survey Water Supply Paper 1541-B. US Geological Survey, Washington, DC.

Wang, S., Fu, B., Piao, S., Lü, Y., Ciais, P., Feng, X., and Wang, Y.: Reduced sediment transport in the Yellow River due to anthropogenic changes, Nature Geoscience, 9, 38, 2016.

---

## Author Response (AR1)

**Letter of Response (hess-2017-261)**

**We would like to thank the Editor for handing the manuscript, and to thanks the Referees for their insightful comments, which have helped improving the manuscript. We provide below our detailed response to each comment. All page and lines numbers refer to the revised marked manuscript.**

**To Reviewer #1:**

**General Comments:**

*It has been a pleasure reading through this contributions. This work characterizes the drought by linking climate anomaly with the change in precipitation-runoff relationship in China's Loess Plateau, and discusses the policy implications of the study to water resource management in a water-limiting environment. The study is scientifically valid, the methods and data sources are well explained, and the results are clear and well presented,  though there are some aspects need to ameliorate. Overall, I would recommend this manuscript for publication in Hydrology and Earth System Sciences, with some comments and suggestions.*

**[Response]**We thank the reviewer for supporting the publication of this MS. The MS has been revised carefully, following the reviewer's comments and suggestions. Our detailed responses follow.

**[Reviewer #1 Comment 1]***Section 2.4.1. Parameters estimation: The paper chooses seven commonly functions as the candidate margins distribution for drought duration and severity, there are some deficiencies in fitting margin distribution function. For example, "by comparison…", I hope the authors can provide quantitative value to determine distribution functions. "drought and severity are fitted with weibull and gamma …", the authors need to show relevant statistical indicators.*

**[Response]**Based on this comment, we provide quantitative values to assess the fit of the marginal distribution functions. We use the root mean square error (RMSE) and the Kolmogorov-Smirnov (K-S) test to select the best-fitting distribution. Table 3 (Page 36) lists the estimated parameters and the results of the goodness-of-fit tests. We find that not all the distributions pass the K-S test at the 95% ($\alpha$=0.05) significance level. Further, considering the RMSE, the Weibull and gamma distributions provide the best-fitting marginal distributions for drought duration and severity, respectively. The results for these distributions are shown in bold and underlined in Table 3.

**[Reviewer #1 Comment 2]***Section 2.4.2. Only the method of Squared Euclidean Distance(SED) is used*

*to perform the goodness-of-fit of joint distribution function, I recommend the authors can adopt more methods to evaluate the fitted copula, such as root mean square error(RMSE), the Akaike information criterion(AIC)...*

**[Response]** We thank the reviewer for this comment. In addition to the squared Euclidean distance (SED) method, we have employed the root mean square error (RMSE) and the Akaike information criterion (AIC) to further evaluate the fitted copula. As shown in Table 4 (Page 37), the Frank copula is the optimal joint distribution function in most watersheds examined in this study, except for the Jialu, Dali and Beiluo watersheds. The optimal goodness-of-fit values obtained using different methods are also shown in bold and underlined.

**[Reviewer #1 Comment 3]***The English expression in this MS is sub-standard; it needs to be improved. The authors should further review the whole paper, although I have pointed some in specific suggestions. In addition, some sentences in the paper are very long, without clear phrasing, so that the reader is sometimes left wondering what the main point of the sentence was. The authors need also notice these problems.*

**[Response]**We have asked a native English-speaking scientist to help us with the language of the revised MS.

**Specific suggestions:**

**[Reviewer #1 Suggestion 1]***Page1.L4, not all readers will know that this re-vegetation is anthropogenic, you need to explicitly state this.*

**[Response]**We explain the details of the revegetation programme. China experienced severe droughts in 1997 and serious floods in 1998, and these events caused serious economic and environmental damage (Tian et al., 2016). In the wake of these disasters, the Chinese government took unprecedented conservation measures (Xu and Cao, 2001), one of which was the Grain for Green Programme (GGP). This initiative was introduced in 1999 to protect the degraded environment (Zhang et al., 1999). The objective of this programme was to convert cropland to plantations and grasslands on steep slopes by compensating farmers with subsidies (Page 3, Lines 17-18 and Page 4, Lines 20-21).

**[Reviewer #1 Suggestion2]***Page1.L5, delete "in the area".*

**[Response]**We have deleted ''in the area'' accordingly. The sentence has been changed to read "*This*

*case study characterizes drought by linking climate anomalies with changes in the precipitation-runoff relationship (PRR) on the Loess Plateau of China, a water-limited region where ongoing revegetation makes drought a major concern.*" (Page 1, Lines 4-6)

**[Reviewer #1 Suggestion3]***Page3.L11,delete "reflect".*

**[Response]**We have deleted the word "reflect" accordingly. The sentence has been changed to read "*Thus, analysing drought characteristics in terms of the changes in the PRR that occur in response to multi-year dry periods is of great importance in estimating the effects of drought and the ecological reconstruction of the Loess Plateau as a whole.*" (Page 3, Lines 25-28)

**[Reviewer #1 Suggestion4]***Page3.L20, as the climate is changing over what years are these long-term averages calculated?*

**[Response]**We state clearly in the revised MS that the long-term averages are calculated for the period of 1960–2000 (Page 4, Lines 6-7).

**[Reviewer #1 Suggestion5]***Page4. L21, "propose use"?*

**[Response]**We have modified this sentence to read "Here, we use the copula function (Shiau, 2006)." (Page 5, Line1)

**[Reviewer #1 Suggestion6]***Page6.L9, states that 7 dry periods are identified yet on Fig 8(a) there are 15 events. This is confusing.*

[Response] "Based on the drought identification method developed in this study, 7 dry periods are identified (including major dry periods and single-year dry periods) on the Loess Plateau as a whole during 1961-1999 (Page 7, Lines 20-22). The purpose of this study is to focus on the changes of the PRR during the major dry periods. Further, considering the variability of the PRR during the dry periods in each watershed (section 3.3), there are 15 dry periods (including significant and non-significant changes) in the 13 studied watersheds, in which the drought regressions fall under the overall regression lines (Page 13, Lines 4-5). We have clarified this point in the revised MS.

**[Reviewer #1 Suggestion7]***Page6.L19, "In1991–1999 (p=0.000) there was a significant decrease change significantly in the PRR", expression is repeated.*

**[Response]**We have modified this sentence to read "However, a significant decrease in the PRR can be identified for 1991–1999 (p=0.000)." (Page 8, Lines 7-8)

**[Reviewer #1 Suggestion8]***Page8.L6, "multi_yeat".*

**[Response]**We have revised "multi_yeat" to "multi_year".

**[Reviewer #1 Suggestion9]***Page8.L10, "Compared to"*

**[Response]**We have revised this sentence to read "*Compared with the annual average precipitation in separate watersheds during 1961–1999, the watersheds where no significant changes in the PRR occurred (Kuye, Dali, Qingjian, Yanhe, and Jinghe) received greater amounts of precipitation.*" (Page 10, Lines 6-7)

**[Reviewer #1 Suggestion10]***Page10.L24, hey you are introducing a new model and a new dataset in the Discussion section. This is very non-standard the structure is all over the place.*

**[Response]** We agree with the reviewer that, in a standard structure, the net primary production (NPP) data that are derived with the terrestrial Carnegie-Ames-Stanford approach (CASA) that we employ in the discussion section on Page 10 L24 should be first explained in section 2.1.

However, we have replaced the NPP data in this section with satellite-derived Leaf Area Index (LAI), according to the comments of Reviewer #2 (see our response to Reviewer #2, general comment 4). In line with this comment, we describe the LAI data in section 2.1 in the revised MS. (Page 4, Lines 17-20)

**[Reviewer #1 Suggestion11]***Fig 5, Precipitation, and many other hydrological variables, have the dimensions of depth / time, and you need to include the time of integration into you units. So your X-axis should have the units of mm/year. When assessing annual trends of annual (or actual E, potential E or Epan) the units are mm/year/year, as in such a plot the X-axisis years, and the Y-axis of an annual P time-series is mm/year, so the slope (or trend)of delta_Y / delta_X has the units of mm/year/year.*

**[Response]**We have revised this figure so that the X-axis and Y-axis represent P (mm/year) and runoff (mm/year), respectively in the revised MS, as shown in Figure 5 (Pages 25-27).

**References**

Tian, F., Feng, X., Zhang, L., Fu, B., Wang, S., Lv, Y., and Wang, P.: Effects of revegetation on soil

moisture under different precipitation gradients in the Loess Plateau, China, Hydrology Research, 2016. nh2016022, 2016.

Xu, J. and Cao, Y.: The socioeconomic impacts and sustainability of the SLCP, Implementing the Natural Forest Protection Program and the Sloping Land Conversion Program: Lessons and Policy Recommendations. CCICED-Task Force on Forests and Grasslands. Beiji     ng: China Forestry Publishing House, 2001.

Zhang, X. P., Zhang, L., McVicar, T. R., Van Niel, T. G., Li, L. T., Li, R., Yang, Q., and Wei, L.: Modelling the impact of afforestation on average annual streamflow in the Loess Plateau, China, Hydrological Processes, 22, 1996-2004, 2008.

**To Reviewer #2:**

**General Comments:**

*The authors analyze the drought impacts on the runoff ratio in China's Loess Plateau. The climate anomaly, relationships between precipitation-runoff, the implications for ecosystem, and the water resource management were discussed in the manuscript. The structure of the manuscript and the problems description are well organized, but there are several serious flaws in the data analysis, methods description, and interpretations of results. Thus, this version of the manuscript can not be accepted for publication in HESS.*

**[Response]**We thank the reviewer for these comments. We have carefully considered all the reviewer's comments, and our responses are shown below. We believe the MS has been substantially improved, and the issues noted by the reviewer have all been addressed.

**[Reviewer #2 Comment 1]***First, the amount of the water consumption for the local communities (domestic and industrial usage) is vital for the runoff ratio in the study period, especially for during the drought. The authors should at least investigate the changes in the water supply for the local communities.*

**[Response]** We agree that the amount of water consumed by the local communities is vital for the runoff ratio, especially during drought periods. For example, Bouwer et al. (2006) concluded that increasing water consumption for irrigation and the degree of runoff variability caused hydropower is three times higher than the variations in runoff under climate change in a densely populated region in the main agricultural irrigation area in India (Page 11, Line 26 to Page 12, Lines 1-3 ). However, the

water consumption for the local communities is not a major issue in our study area, which is composed of 13 hilly catchments on the Loess Plateau.

The catchments examined in our study lie in the part of the Loess Plateau with the greatest relief, and the Mu Us Desert is located in the northwest and the Weihe Plain is located in the southeast (Page 11, Lines 23-25). The water consumed by the local communities on the Loess Plateau is fed to the residential areas, which are mainly located in the flat areas at the outlets of the catchments (the distribution of residential areas is shown in Fig. 1, Page 21). Moreover, the population shows a tendency to move from the catchment area to the major cities, which are located along the mainstream of the river basin in the Weihe Plain (i.e., Baoji, Xi'an, these cities contain 57.35% of the population within the study area) because of the accelerated urbanization that has taken place in this area since the 1980s (Hu et al., 2001). (Page 12, Lines 5-9)

In line with these considerations, in the studies of runoff variability of the same catchments, the water supply to the local communities is also not included as a factor influencing runoff. Instead, the anthropogenic factors that drive the changes in runoff in the 13 studied catchments include terrace building and soil conservation measures (Wang et al., 2009; Shi et al., 2013; Chang et al., 2015) (Page 12, Lines 14-17). We have added explanations of this point to the revised MS.

**[Reviewer #2 Comment 2]** *The precipitation-runoff relationships can be influenced by the land use, surface water diversion, irrigation scheme, groundwater abstraction, and the water storage in the(sub) catchment. These issues should be addressed for identifying the influence of drought on the water yield.*

**[Response]** We agree with the reviewer that the precipitation-runoff relationships can be influenced by factors other than climate conditions. We have carefully considered each possible factor in our study, as described below.

The catchments lie in the part of the Loess Plateau with the greatest relief, and the vegetation in the catchments is mostly rain-fed. Thus, the effects of irrigation schemes within the study area can be neglected (Feng et al., 2016) (Page 11, Lines 23-26). The thickness of the loess within the catchments is greater than 100 m (Derbyshire et al., 1998), and the groundwater is minimally impacted by the surface eco-hydrological processes; thus, groundwater recharge and groundwater discharge are not considered in the study area (Page 11, Lines 26-29). Finally, any diversions of surface water and water storage are found in the residential areas at the outlets of these catchments  Therefore, their effects are not

included as potential impact factors affecting the precipitation-runoff relationships (Page 12, Lines 10-12).

However, soil conservation measures, including the construction of terraces and sediment-trapping dams, have been implemented in the Loess Plateau since the 1950s (Wang et al., 2016) (Page 12, Lines 14-15). We have added a description of the influence of these human-stimulated effects on precipitation-runoff relationships in the revised MS. The partial correlation method is used to isolate the impacts of anthropogenic influences from climate-related factors (Page 12, Lines 17-18).

For the entire period of 1982–1999, the runoff displays a decreasing trend (Fig. 8, Page 30 ).Terrace construction played an important role in producing the reduction in the runoff ratio from the 1980s to the 1990s ($p=0.048$, Fig. 9, Page 31). The effects of other anthropogenic activities, including dam construction, tree plantations and pastures, did not cause the observed change in the runoff ratio in this period. Terrace construction contributed 25% of the reduction in the runoff ratio in the 1990s. Thus, drought events are the major factor driving the reduction in runoff in the study area (Page 12, Lines 20-23). We have added a description of this analysis to the revised MS.

**[Reviewer #2 Comment 3]***Section 2.2 The proposed classification method of drought events, drought periods, the interpretations of results, and the upscale processes from 13 sub catchments to regional precipitation anomaly are not clear enough to support the publication of this version of the manuscript in HESS.*

**[Response]**We thank the reviewer for this comment. We have rewritten section 2.2 to clarify the classification method. The relevant text reads as follows in the revised MS.

"I*n this study, we define drought based on annual precipitation for two aspects. On the one hand, the amount of precipitation is the most important climatic control of drought conditions (Mishra and Singh, 2010). Moreover, because we are interested in determining whether the runoff response differs for multiyear droughts, we do not consider runoff in identifying drought events.*

*We first calculate the precipitation anomaly (PA) values in the studied watersheds on the Loess Plateau. The time series of anomaly values are divided by the mean annual precipitation and smoothed with a 3-year moving average. Positive PA values indicate that the observed precipitation is higher than the median. On the other hand, negative PA values indicate that the observed precipitation is below the median and imply the possible occurrence of a drought. Each drought event is characterized in terms of*

*its duration and severity. Studies have shown that the drought events with shorter durations but greater intensities or lower intensities but greater durations cause serious water-supply and other drought-related problems (Shiau, 2006; Naresh et al., 2009). Therefore, the basic rules for identifying drought events in this study are (1) a PA value for a single year of $\leqslant$ -10% or (2) mean PA values of less than 0 for more than three consecutive years. Note that the PA value of the starting year of each drought period is negative.*

*In this study, the cumulative PA values during each drought period are used to measure drought severity (for convenience, drought severity is multiplied by -1 to obtain a positive value). Based on the rules mentioned above, we identified all of the drought events that occurred in each watershed. To reflect the response of the PRR to drought events over the years, we must ensure that the dry periods are sufficiently long and severe. In the subsequent analysis, we consider only drought events with durations $\geqslant$ 5years and mean annual PA values $\leqslant$ -5% during the drought period. Finally, the dry events are classified into major dry period and single-year dry period.*

*We use the Kolmogorov-Smirnov (K-S) (Massey, 1951) test to determine whether annual precipitation and runoff data follow a roughly normal distribution. A Box-Cox transformation is applied to those data that are not normally distributed (Box and Cox, 1964). After identifying the major drought events, we examine whether the change in the PRR is statistically significant compared to the historic record using Student's t-test (p $\leqslant$ 0.05). The historical records refer to scatterplots of annual precipitation-runoff during the period of 1961–1999, except for particular major drought periods. For example, when the drought that occurred in 1970–1974 is considered, the corresponding historical record includes a precipitation-runoff scatter plot that includes data from 1961–1969 and 1975–1999. For the drought that occurred in 1991–1999, the corresponding historical record refers to a precipitation-runoff scatterplots containing data from 1961–1990.*"

**[Reviewer #2 Comment 4]***The NPP estimation based on the remote sensing data (2000-2008) could not support the analysis results of the drought on the ecosystem from 1961 to 1999. The authors need to find at least the data in one of the main drought period defined in this manuscript and another normal period to illustrate the difference for determining the drought impacts.*

**[Response]**We agree with the reviewer that we need to examine data for both drought periods and normal periods to illustrate the impacts of drought. Due to the lack of NPP data before 1999, we have used the AVHRR GIMMS LAI3g data, which covers the period from 1982 to 1999 (Page 4, Lines

17-18). We choose the drought period of 1991−1999 as an example, and we find that the LAI decreases significantly ($p=0.032$, Student's t test) in 1991−1999 compared to 1984−1990 (Fig. 11, Page 33; Page 13, Lines 2-3). We have included this new analysis in the revised MS.

**[Reviewer #2 Comment 5]***The English should be substantial improved to a certain level that the readers can not misunderstand the correct information.*

**[Response]**We have asked a native English-speaking scientist to help us to revise the language in the MS.

**Specific comments:**

**[Reviewer #2Comment 1]***Affiliation: Shaanxi? should be Shanxi.*

**[Response]**The correct affiliation is Shaanxi. Shanxi is a different province in China, which is not related to this MS.

**[Reviewer #2Comment 2]***Page 1, line 1, "is" should be "are".*

**[Response]**This sentence has been changed to read "*The frequency and intensity of drought are increasing dramatically as global warming progresses.*" (Page 1, Line 1)

**[Reviewer #2Comment 3]***Page 1, line 5, only the re-vegetation that makes the drought a major concern?*

**[Response]**The vegetation restoration programme in China represents the largest investment that has been made to restore the ecosystem in this developing country. Given the limited water resources on the Loess Plateau, the sustainability of vegetation restoration programs is a major concern of scientific research and policy makers there (Feng et al., 2016) (Page 13, Lines 18-20). We have clarified this point in the revised MS.

**[Reviewer #2Comment 4]***Page 1, line 12 delete the "around" after "precipitation"*

**[Response]**We have deleted the word " around " where it follows " precipitation".

**[Reviewer #2Comment 5]***Page 1, line 13-14,"NPP" and "PRR" should not be abbreviation in first appearance.*

**[Response]**We have changed this sentence to read "*At the same time, multiyear drought events also lead to significant changes in the leaf area index (LAI).*" (Page 13, Lines 18-20). Here, NPP has been replaced with LAI. A detailed explanation of this change can found in the response to general comment 4 of Reviewer #2.

**[Reviewer #2Comment 6]***page 2, line 9-11, weird sentence.*

**[Response]** We are sorry that this part of the text was unclear. The sentence has been changed to read "*For example, the soil moisture indicator (Xia et al., 2014), the crop drought indicator (Duff et al.,1997) and the crop water demand indicators are used to identify agricultural drought events, which are periods that feature dry soil conditions and result from below-average precipitation, intense but less frequent rain events, or above-normal evaporation. All of these factors lead to reductions in crop production and plant growth.*" (Page 2, Lines 14-18)

**[Reviewer #2Comment 7]***Page 2, line 30, replace the "with" with "by".*

**[Response]**We have revised the sentence to read "*Therefore, the shift in the PPR caused by an extended drought will eventually have an adverse effect on the ecosystem service of water yield.*" (Page 3, Lines 10-11)

**[Reviewer #2Comment 8]***Page3, line 25, please indicate the data length or periods.*

**[Response]**In the revised MS, we have indicated that the data extend from 1961 to 1999 (Page 4, Lines 14-15).

**[Reviewer #2Comment 9]***Page 3, line 27, website in the bracket does not match the text.*

**[Response]**According to this comment, we have revised the website in the brackets to http://www.yellowriver.gov.cn/.

**[Reviewer #2Comment 10]***Page 4, line 2, replace "its" with "in".*

**[Response]** Following this comment, we have modified the sentence to read "*We first calculate the precipitation anomaly (PA) values in the studied watersheds on the Loess Plateau. The time series of anomaly values are divided by the mean annual precipitation and smoothed with a 3-year moving average.*" (Page 4, Lines 29-30)

**[Reviewer #2Comment 11]***Page 4, line4, conditions 2 should be page 6, clarified.*

**[Response]**We have checked condition 2 on Page 4, line 4 in the MS. As one of the basic rules for identifying drought events in this study, we think putting it in section 2.2 ((Page 5, Lines 7-8) is more appropriate.

**[Reviewer #2Comment 12]***Page 6, line 21, please indentify the time period for "long term".*

**[Response]** In the revised MS, we have clarified that the phrase "long term" refers to the period from1961 to 1990 (Page 8, Lines 9-10).

**[Reviewer #2Comment 13]***Page 7, line 25-27, long sentence.*

**[Response]**This sentence on Page 7, line 25-27 has been revised to read "*The return period of the drought period that occurred in 1970-1974 is approximately 5.74 years, given the corresponding drought characteristics. Therefore, the next drought event similar to the drought period that took place from 1970 to 1974 occurred around 1980. In 1979–1983, the drought duration reached 5 years, which is close to the estimated return period.*" (Page 9, Lines 17-21)

**[Reviewer #2Comment 14] section 3***.3, please re-write the first paragraph.*

**[Response]**We have rewritten the first paragraph of section 3.3. It now reads as follows.

"*Prolonged multi-year drought causes significant damages in natural environments. Fig. 5 demonstrates the range of changes in the PRR under sustained precipitation decreases. According to the direction of change, the dry period regression line is mainly located above or below the overall regression line, and the PRR in the 13 studied watersheds exhibits no significant change when the regression line of the dry period is above the total regression line. Of the 15 cases in which dry events fell under the overall regression line in the 13 watersheds from 1961 to 1999, significant changes in the PRR (p < 0.05) occurred in 9 cases, accounting for approximately 60% of the total cases. In these cases, the dry period regression line lies lower than nearly all of the other parts of the historical record, indicating unprecedentedly low runoff generation rates for the given precipitation values. Thus, we conclude that reduced runoff occurs in years with decreased precipitation due to the reduction in precipitation; moreover, less runoff than expected occurs during multi-year drought periods.*"

**[Reviewer #2Comment 15]***Page 8, line 10, where are the basins with significant changes in precipitation in table 1?*

**[Response]** We apologize for the confusion. The phrase " significant changes " on Page 8, line 10 refers to those watersheds which display significant changes in the PRR. Comparing the annual mean precipitation in separate watersheds during 1961–1999 (Table 1, Page 34), we find that multiyear drought events are more likely to cause significant changes in the PRR of basins that receive less precipitation. We have clarified this point in the revised MS.

**[Reviewer #2Comment 16]***Page 9, line 5, replace the "as well as " with "and "*

**[Response]**We have modified this sentence to read "*Prolonged multi-year drought events cause significant damages to both the natural environment and the development of human societies (Belal* et al*., 2014).*" (Page 11, Lines 5-6)

**[Reviewer #2Comment 17]***Page 9, line11, should be "http://www.mwr.gov.cn"*
**[Response]**We have revised this link to "http://www.mwr.gov.cn".

**[Reviewer #2Comment 18]***Figure 1, where is the Yellow river? it is indicated on the up-left small plot that the Yellow river flows through the loess plateau.*
**[Response]**The Yellow River flows through the Loess Plateau. We have labelled the Yellow River in the inset map in the upper left-hand corner of Fig. 1 (Page 21).

**[Reviewer #2Comment 19]***Figure 2, Do you use the average of rainfall for the 13 watersheds? The description of drought events for condition 1 and 2in section 2.2 may not be applied on the year 1974, when the 3- year moving average should be lowest in the first main drought period. But the 3-year moving average in1970 in the figure is lowest.*
**[Response]**Yes, we used the average rainfall for the 13 watersheds in Figure 2. After re-examining the calculation of the original data, we found a mistake in our computation of the 3-year moving average. After revising the results, the first main drought period is shown to have occurred in 1970–1974. As shown in Fig. 2 (Page 22), the rainfall anomaly in 1974 is -23%, which is consistent with condition 1 in section 2.2. In the first main drought period, the 3-year moving average of 1974 is only smaller than in 1972. We have corrected this mistake in the revised MS.

**[Reviewer #2Comment 20]***Figure 3, What are the historical records? Apparently, the historical records in three plots are different, why? better to use the same scale for x-axis in three plots.*

**[Response]**The historical records in Figure 3 refer to scatterplots of annual precipitation-runoff during the period of 1961−1999, except for particular major drought periods. For example, when the drought that occurred in 1970−1974 is considered, the corresponding historical record includes a precipitation-runoff scatter plot that includes data from 1961-1969 and 1975−1999. For the drought that occurred in 1991−1999, the corresponding historical record refers to a precipitation-runoff scatterplot containing data from 1961−1990 (Page 5, Lines 23-28). Because three different multiyear dry periods occurred on the Loess Plateau during 1961−1999, the corresponding historical records are different in the three plots. We have clarified this point in the revised MS.

**[Reviewer #2Comment 21]**_Figure 5, the drought periods in different sub-catchments are not identical, why? again, what are these historical records?_

**[Response]**The drought periods vary spatially, as shown in Figure 5. The drought period in each catchment is identified using the local precipitation data and changes in the precipitation-runoff relationship.

As in Figure 3, the historical records refer to the annual precipitation-runoff scatter plot for the period of 1961−1999, except for particular major drought periods. A detailed explanation is provided in the response to comment 20 of Reviewer #2.

**[Reviewer #2Comment 22]**_Figure 7, what is the drought event corresponding to the return period in figure 7d?_

**[Response]**The drought event corresponding to the return period in Fig. 7d has a drought duration and severity that caused a significant change in the PRR in 8 of the studied watersheds (Page 10, Lines 22-25). We have clarified this point in the revised MS.

**[Reviewer #2Comment 23]**_Figure 8, at least show the whole legend of Figure 8a. is it the average return period of the drought events in 8b?_

**[Response]**Based on this comment, the legend of Fig. 10 (Page 32) has been revised to read " _(a) Characteristics of drought events with significant changes and (b) spatial distribution of the joint return period(four regions)._" It is the average return period of the drought events in Fig. 10(b).

**[Reviewer #2Comment 24]**_Figure 9, add "change" after the significant in the caption._

**[Response]** The Figure 9 had been deleted in the revised manuscript.

[revised manuscript text omitted]
 | 0.204 | 4.974 | 0.265 | 0.197 | 5.642 | 0.288 | 0.214 | 5.113 | 0.301 | **0.189** | **4.312** | **0.249** | 0.190 | 5.230 | 0.254 |